# East European sedimentary basins long heated by a fading mantle upwelling

Alik Ismail-Zadeh [1] ✉, Anne Davaille[2], Jean Besse[3] & Yuri Volozh[4]

A strong negative anomaly of seismic wave velocities at the core-mantle boundary (the Perm Anomaly) beneath the East European platform is attributed to the remnant of a deep mantle upwelling. The interaction between the upwelling and the East European lithosphere in the geological past and its resulting surface manifestations are still poorly understood. Using mantle plume modelling and global plate motion reconstructions, we show here that the East European lithosphere is likely to have been situated over the weakening Perm Anomaly upwelling for about 150–200 million years. As the East European platform moved above the Perm Anomaly in post-Jurassic times, the vertical tectonic movements recorded in sedimentary hydrocarbon-rich basins show either hiatus/uplift or insignificant subsidence. Analytical modelling of heat conduction through the lithosphere demonstrates that the basins have been slowly heated for a long time by the Perm Anomaly upwelling, creating suitable conditions for hydrocarbon maturation. This suggests a profound relationship between mantle plume dynamics, basin evolution, and hydrocarbon generation.

Seismic tomography reveals the presence of a province of low seismic shear-wave velocity at the core-mantle boundary beneath the southeastern part of the East European (EE) platform, called the Perm Anomaly (PA)[1,2] (Fig. 1). The PA extends radially to an estimated 400–600 km above the core-mantle boundary[3]. It is separated from the two large provinces of negative seismic wave velocity anomalies below the Pacific and the Indo-Atlantic regions[4–6] by the Neo-Tethys subduction to the south, the Pacific subduction to the east and north, and the Mongol-Okhotsk subduction to the north (Fig. 1). Geologic, paleomagnetic, and seismic studies show that the Neo-Tethys and Pacific subductions are long-lived features that have been active at least for the last 300 million years (My)[7,8], delivering a nearly continuous downwelling flow of cold material on the core-mantle boundary around PA. The longevity of these downwellings and mass conservation in the mantle imply the longevity of hot upwellings there as well. While the PA's origin is still debated, PA was first linked to a thermally prominent mantle plume[1], which was active in the Permian-Triassic times and responsible for the Siberian trap formation[9,10] (see section

SD1 of the Supplementary Information (SI)). Under the oceanic lithosphere, the subsequent whereabouts of such a plume would be recorded by a chain of volcanoes. Instead, the EE platform displays sedimentary basins rich in hydrocarbons, but no obvious volcanic trail. Despite significant progress made in understanding the thermal evolution of the continental lithosphere[11] as well as sedimentary basins and hydrocarbon generation[12,13], the links between deep mantle heat sources and hydrocarbon maturation are still poorly understood.

So, in this paper, we address the following key question: could a mantle upwelling (or mantle plume), even though it was weakening thermally with time, have influenced the evolution of sedimentary basins and hydrocarbon maturation within the EE platform? To answer this question, we (i) show that PA can be considered as the remnant part of a fading mantle plume, (ii) study the PA upwelling track on the EE platform since the Early Jurassic times, (iii) examine the subsidence history of the sedimentary basins along the PA upwelling track, and finally, (iv) analyse the thermal regimes associated with heat conduction through the lithosphere and relate them to hydrocarbon

[1]Karlsruhe Institute of Technology, Institute of Applied Geosciences, Karlsruhe, Germany. [2]Laboratoire FAST, CNRS and Université Paris-Saclay, Orsay, France. [3]Université de Paris Cité, Institut de Physique du Globe de Paris, CNRS, Paris, France. [4]MOSESTRO Exploration, Tel Aviv, Israel. ✉ e-mail: alik.ismail-zadeh@kit.edu

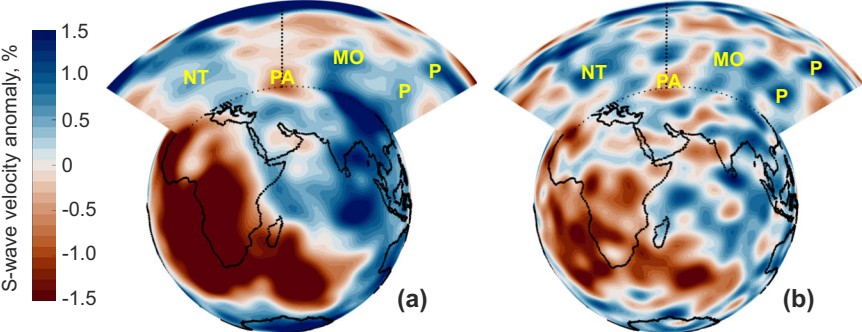

**Fig. 1 | Structure of the mantle below Eurasia as seen by seismic S-wave velocity heterogeneities dVs/Vs.** Negative velocity anomalies are usually interpreted as the presence of hotter material, while positive velocity anomalies as the signature of colder material. The present position of the continents (black lines) is superimposed on the SMEAN seismic tomography model (panel **a**, ref. 4), which is an average of three different S-wave tomography models, and on the SEMUCB-WM1 S-wave tomography model (panel **b**, ref. 72) at the depth of 2800 km, along with radial cross-sections. The lower velocity Perm Anomaly (PA) province is surrounded by faster velocity areas corresponding to the Neo-Tethys (NT), Pacific (P)[7] and Mongol-Okhotsk (MO)[8] subduction zones. The colormap is generated based on the work by Crameri et al.[73].

generation and maturation in the EE basins that have been heated over the thermally fading mantle plume since the Jurassic times.

## Results

### Perm anomaly as a remnant part of a fading mantle plume

The Earth's surface and its evolution are shaped by the thermal convective motions inside its mantle. Lithospheric plates descending into the mantle constitute the upper cold part of this convective system, and mantle plumes are their hot counterparts. The plumes are likely to be responsible for hotspot volcanism[14], formation of large igneous provinces (LIP)[5,9,15], continental stretching and break-up[15], mass extinction[16], and eruption of diamond bearing kimberlites[17]. Depending on their temperature, composition, and rheology, a wide range of morphology and temporal behaviour are expected for mantle upwellings[6,18,19].

Mantle plumes are generated from the hot thermal boundary layer (TBL), which is produced by conductive heating of the material at the core-mantle boundary. When the TBL is thick enough, it becomes gravitationally unstable, generating upwellings (i.e., plumes) that drain locally the TBL[19,20]. The latter is replaced by cold material (such as remnants of cold subducted slabs) that will take time to reheat. A plume is therefore a transient feature. While the discrimination of low-velocity anomalies (seen in seismic tomography models) in active and less active plumes is a challenging problem, laboratory and numerical experiments provide evidence of thermal plumes fading with time[20,21]. They show that: (i) the mean velocity of the fluid within a plume decreases and hence thermal diffusion becomes a major agent in the heat transfer as the plume grows older; (ii) the plume tail (or conduit) begins to disappear from its bottom-half up (Fig. 2a), leaving at the end only the cooling and shrinking sub-lithospheric overhang and the plume foot; and (iii) the plume life-span can reach 200–300 My for typical lower mantle viscosity.

Figure 2 presents several stages of the PA mantle plume thermal fading from the Early Jurassic times (*b*) to the present (*f*) obtained by numerical modelling of the mantle plume diffusive decay (see *Methods* for the model's detail). We assume that the PA plume was still thermally prominent in the Early Jurassic (Fig. 2b), although the plume tail was diminishing in size due to thermal diffusion (Fig. 2c) to get cut from the lower mantle TBL in the Late Cretaceous times (Fig. 2d) and to become invisible at least since the Late Paleogene (Fig. 2e, f). The remnant foot of the modelled mantle plume may explain the PA presence at the core-mantle boundary[1] (Fig. 2g).

The paleogeographic and palinspastic reconstruction of the motion of the northern Eurasia based on palaeomagnetism[22] shows that the EE platform moved north-eastward at the average rate of about 6 cm yr$^{-1}$ in the Middle to Late Triassic times. The buoyant plume

material would have been advected by the lithosphere motion, while spreading in the lateral direction under its own buoyancy[23]. The relatively rapid lithosphere movement may therefore have enhanced the thinning of the plume head in the direction of movement and its subsequent cooling. Meanwhile, plume heads are likely to be thermally fading within a short time (a few My) if significant melting occurs during mantle upwelling. The melting could produce massive volcanic eruptions and the development of LIP[24]. As the present numerical model does not consider melting processes, the modelled mantle plume head is preserved for a longer time. Nevertheless, our model shows that thermal diffusion plays a role in the fate of advecting mantle plumes[21] and may provide a powerful explanation of PA at the core-mantle boundary.

### East European lithosphere above the Perm Anomaly plume: implication for the evolution of sedimentary basins

The continental lithosphere is in a state of permanent relative movements due to the basal traction generated by mantle flow, oceanic slab pull, and mid-ocean ridge push as main driving forces of plate tectonics[25]. To determine the position of Eurasia during the past 200 My, we analyse its movement using three models of global plate motion (GPM)[26–28] (hereinafter the models are referred to as Se2012, Mü2016, and Ma2016, respectively; see *Methods* as well as SD3 of SI).

The projection of the PA fading plume on the Earth's surface has been traced for the past 200 My (Fig. 3), starting from the present position of the plume's centre (51°N, 52°E). In a framework where the PA plume remains fixed on the core-mantle boundary, the EE platform has been 'dancing' above the PA plume at least since the Early Cretaceous times (-150 My ago, Ma). A close inspection of the three modelled tracks of the projection of the PA plume's centre (Fig. 3) shows that the tracks are close to each other from the present to the Late Cretaceous times (-100 Ma). Starting from the Early Cretaceous, the tracks diverge to the northeast, to the northwest, and to the southeast according to Se2012, Ma2016, and Mü2016 models, respectively. However, despite the tracks' divergence, all three track models support our major finding that the EE platform has been situated above the PA fading plume for about 150–200 My.

The track models also suggest that the sedimentary basins of the EE platform have been situated above PA for a long geological time. As the Pricaspian basin moved over PA (according to the SE2012 and Mü2016 models), the basin's area underwent significant uplift recorded in sediments as a hiatus lasting from -208 Ma to 170 Ma[29] (Fig. 4). We attribute this uplift to the PA mantle plume impinging under the EE lithosphere in the Late Triassic–Early Jurassic, at times when the upwelling would be still connected to the deep mantle and powerful

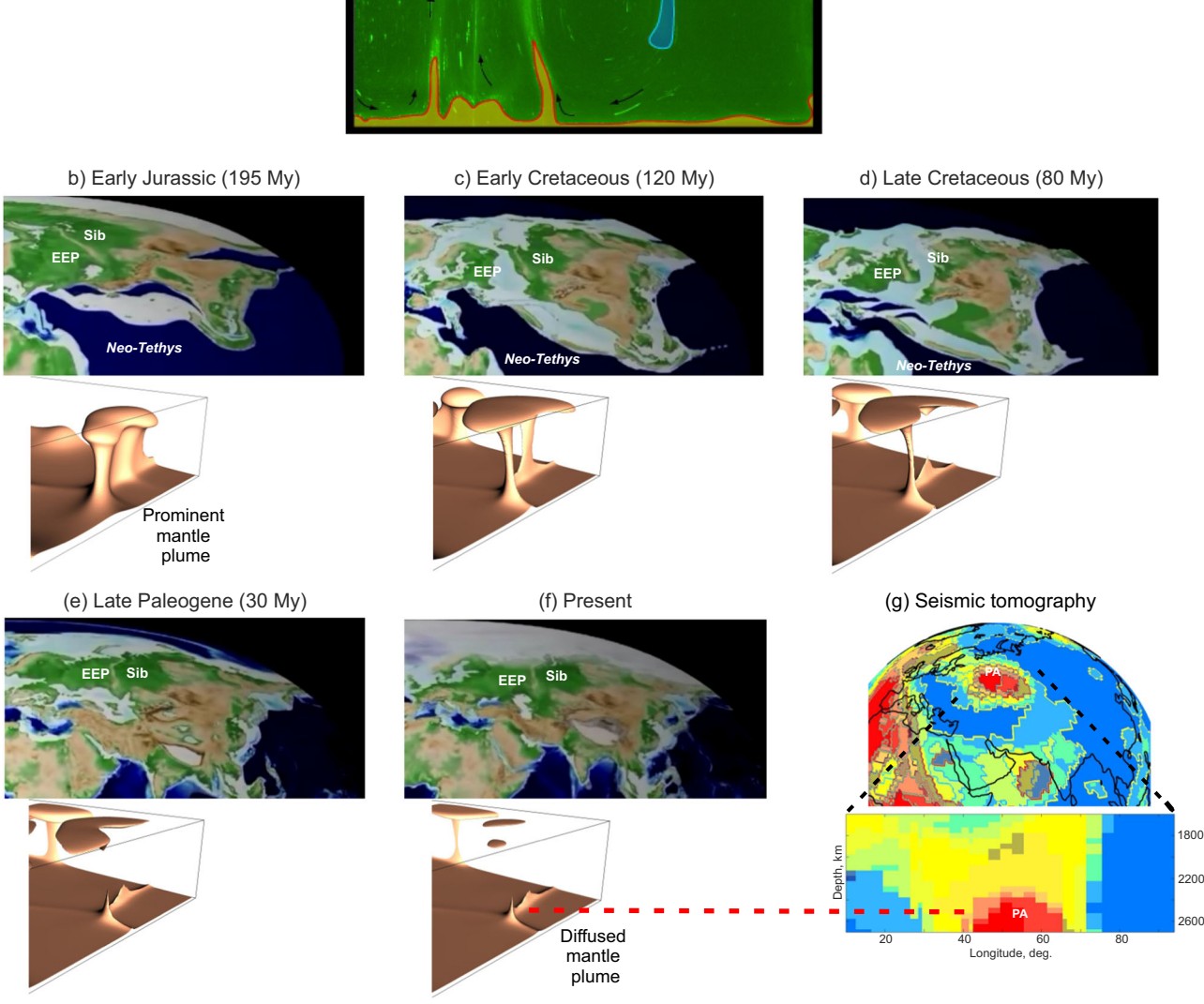

**Fig. 2 | A schematic evolution of the mantle plume associated with the Perm Anomaly beneath the East European platform since the Early Jurassic times.** Panel **a** illustrates a snapshot of the laboratory experiment on thermal plume development and fading[20]. The blue shading indicates the colder regions (T < 23 °C) and the orange shading the hotter regions (T > 39 °C). Panels **b**–**f**: plate tectonic reconstructions[74] (in the upper part of each panel) are supplemented by snapshots of the isothermal surface (T = 3000 K) of the thermally decaying plume predicted by a three-dimensional numerical model of the mantle plume diffusion[21] (in the lower part of the panel). Panel **g** presents two vote maps obtained by cluster analysis of five recent S-waves tomographic models[3], where the deep blue represents areas where all the seismic tomography models found fast Vs-velocity, and the bright red the areas where all the models found slow Vs-velocities. They show that PA is recovered by all models just above the core-mantle boundary beneath EEP (red area on the top map). The dashed red line shows a potential link between the Perm Anomaly and a diffused mantle plume. PA the Perm Anomaly, EEP the East European Platform and Sib Siberia.

enough (Fig. 2b) to generate a prolonged uplift during the Early Jurassic and then a slow subsidence of the Pricaspian basin since the Middle Jurassic times (Fig. 4).

The tectonic subsidence analysis (see *Methods* and SD4 of SI) based on borehole data, such as sediment lithology, thickness, porosity, and age, which have been collected in the EE basins (see Fig. 3 for the borehole locations), shows no or insignificant subsidence during the past 150 My (Fig. 4). Particularly, the Moscow basin became an area of erosion at least since the Early Cretaceous[30] (~150–140 Ma) and the Timan-Pechora basin since the Late Cretaceous/Paleogene times[31]. The Pre-Uralian foredeep is expressed as a series of major depressions extending from the Timan-Pechora basin via the Volga-Ural basin towards the Pricaspian basin and shows no (or little) subsidence taking place for more than a hundred million years.

Magmatic events can indicate how strong thermal perturbations have been in the mantle. The passage over the PA plume is likely to have generated tholeiitic and calc-alkali basalt eruptions in the southwestern part of West Siberia in the Late Triassic times[32,33]. However, no track of significant volcanism has been recorded on the surface of the EE lithosphere since the Early Jurassic times[34]. The absence of regional volcanism is likely to be associated with the decay of the PA plume and its inability to generate significant melting, while

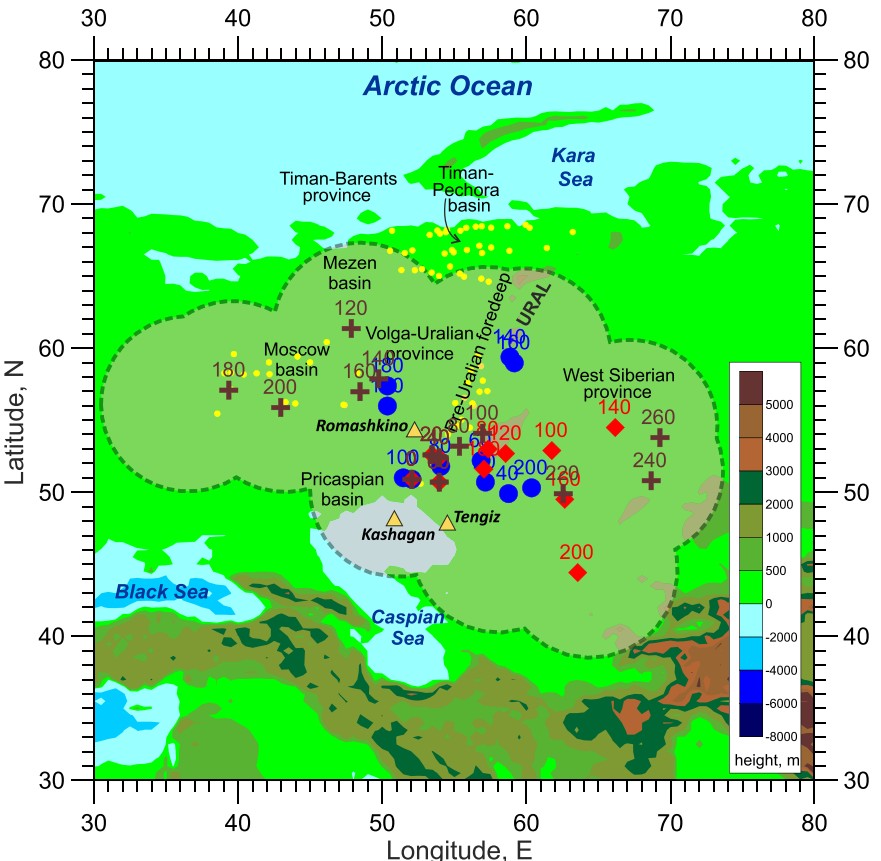

**Fig. 3 | Geographical setting of the studied area presenting tracks of the projection of the PA plume's centre on the surface of western Eurasia from the present position of this centre (51°N, 52°E) to its position in the Early Jurassic (~200 Ma).** The tracks are derived from three GPM models: Se2012 (blue dot symbols), Mü2016 (red diamond symbols), and Ma2016 (brown cross symbols) (see SD3 of SI for details on the GPM models). The symbols mark the position of the related model projection of the PA plume's centre on the surface at the time indicated above the symbol (in Ma). According to the Ma2016 model, the PA plume was located beneath the southern part of the West Siberian Basin between 240 Ma and 260 Ma and likely to be responsible for the Permo-Triassic volcanism found in drilling and attributed to a plume[10] (see Supplementary Fig. S5 of SI) The bold dashed curve delineates a region comprising the 12°-diameter circles around the modelled tracks, representing the spatial uncertainties of the GPM models as well as the typical extent of the lowest part of the seismic wave anomaly on the core-mantle boundary. Yellow dot symbols mark the location of the drilled boreholes used in this study, and yellow triangles are the locations of giant oil/gas fields.

the thick (more than 150 km; refs. 35–37; see SD2 and Supplementary Figs. S2–S4 of SI) EE lithosphere has prevented melts to reach the surface. A study on the Australian continent has shown[38] that melting is likely to occur along hotspot tracks, where the lithosphere is less than 150 km thick.

Thermal evolution of the continental lithosphere and sedimentary basins has been explained by a model of instantaneous heating[39]. According to this model, rapid initial stretching of the lithosphere due to a mantle plume impact and the related crustal thinning are accompanied by an isostatic subsidence of its surface. The instantaneous heating of the lithosphere results in melt generation, its segregation and penetration into the uppermost lithosphere causing magmatism. The impact of a mantle upwelling normally lasts several million years, depending on the continental plate movement above the plume. The subsequent thermal cooling of the lithosphere as it moves away from the mantle plume and erosion of uplifted sediments result in a slow subsidence and sedimentary basin evolution. However, the PA plume impact on the EE lithosphere was different.

We consider that when the EE platform moved over the PA plume, the plume became thermally weaker and unable to significantly stretch the thick EE lithosphere. However, the plume could support the lithosphere beneath the Pricaspian basin to generate the 38-My hiatus in regional sedimentation. Although not hot enough anymore to produce major volcanism, this plume could keep warm the EE lithosphere for a long geological time and prevent tectonic subsidence (Fig. 4).

Conductive heating of the lithosphere by an underlying mantle upwelling is a geologically slow process, but its thermal effects are not negligible when the plume heats the lithosphere for more than a hundred million years. Here we analyse the contribution of the mantle plume's conductive heating on the thermal evolution of the lithosphere and sedimentary basins, which were initially in a steady thermal state. We employ the plate heating model of the lithosphere[11] assuming that the interaction between the plume and the base of the lithosphere started 200 Ma (see *Methods* for the model detail).

As the present thickness of the EE lithosphere ranges from about 150 km to about 200 km (refs. 35–37,40), we consider these two values for the lithosphere thickness in the modelling. After the bottom temperature of the 150-km thick lithosphere ($T_l$ = 1283 K) was perturbed by the hot PA plume in the Late Triassic, the surface heat flow has gained in 200 My up to 9 mW m$^{-2}$ and 12 mW m$^{-2}$ for plume excess temperatures of 487 deg. and 687 deg., respectively (Fig. 5a, blue lines). For the thicker lithosphere (200 km) with a bottom temperature $T_l$ = 1610K, the surface heat flow has gained in 200 My up to 3.1 mW m$^{-2}$ for a plume excess temperature of 360 deg. (Fig. 5a, dashed red line). The model results show that in the case of a thick lithosphere (150–200 km), the temperature gain within 200 My at depths of 6 km to 10 km ranges between about 3 deg. and 40 deg. depending on the plume excess temperature (Fig. 5b, c).

We note that the model results are valid assuming the lithosphere basal temperature to be constant. However, as the PA plume

became thermally weaker with time: its head shrunk (Fig. 2) with a lower heat flux to the surface, and temperature at the base of the lithosphere decreased. Hence, the modelled temperature and heat flow gains should be considered as the upper limits to the crustal temperature gain and its heat flow. Considering the temperature gain to decrease with time over 200 My due to the PA plume's thermal decay, the temperature increase will be not enough to generate plume-impinged rifting[41].

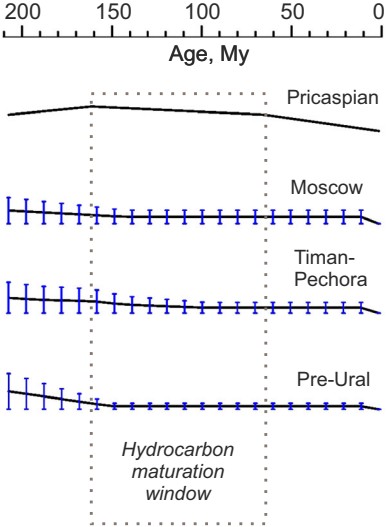

**Fig. 4 | Curves of normalised vertical tectonic movement in the sedimentary basins of the EE platform derived from backstripping of sediments.** Vertical short lines on the curves present the maximum variations in the normalised vertical movements (see *Methods* for more detail about the backstripping and normalisation procedures). The hydrocarbon maturation window is marked by a dotted line (see Table 1 for the maturation times and the estimated hydrocarbon resources).

So, the additional heating due to the presence of the plume beneath the lithosphere for about 150–200 My may explain the lack of basin subsidence observed in the central and eastern parts of the EE platform. Moreover, dikes and sills of basalts could potentially penetrate the thick lithosphere up to intermediate depths providing an additional heat source to the crust.

## Discussion

As heat comes from the mantle and the crust, it warms sedimentary basins above, and hence influences the maturation of hydrocarbons. The temperature required to alter organic material can be reached during a gradual burial and depends on the thermal conditions in the area of burial. At some interval of temperatures and pressures[42], depending on how rapid the source rock has been buried and heated, kerogens (organic compounds) release hydrocarbons (oil or gas). Analyses of hydrocarbon maturity are based on the degree of thermal degradation of kerogens after sedimentary burial and on the reflectivity of vitrinite, a primary component of kerogens[43]. Sedimentary organic matter buried in a basin is heated up until it reaches the oil window (333 K–473 K). The thermal history of sedimentary basins can be retrieved from the knowledge of the present temperature measured in boreholes combined with models of geodynamic and structural evolution of the basins[44]. The subsidence history provides constraints on depths (and pressures) of source rock layers in sedimentary basins. Thermal modelling of the basins allows determining temperatures in the layers in the past.

As the EE basins did not experience significant subsidence since the Cretaceous times, the organic matter was heated at almost the same depth until thermal degradation of kerogens and generation of oil and gas (see Table 1 for the estimated times of hydrocarbon maturation in the EE basins). With the absence of significant subsidence, the only way to bring the source rocks to the oil window is to increase the rock's temperature. At the geotherm used in the modelling (see *Methods*), the crustal temperature at depths of 6 km to 10 km will range from 339 K to 366 K. A mantle plume beneath the thick lithosphere could heat sediments at a depth of 10 km from about

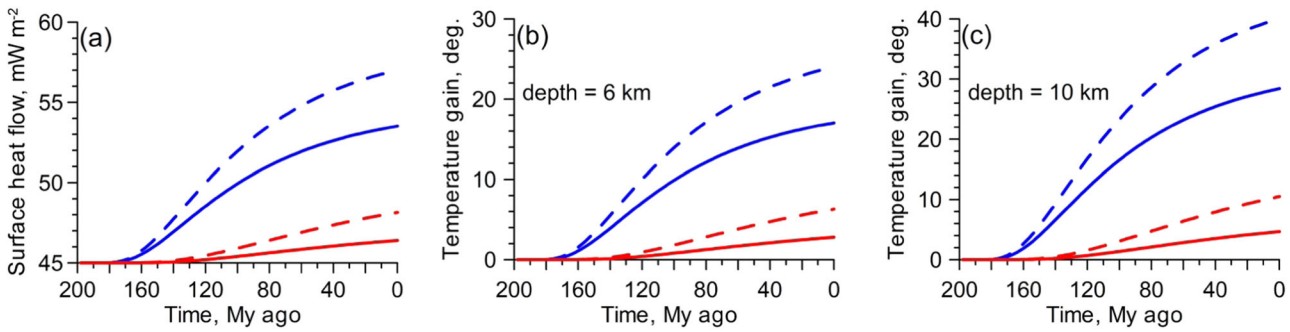

**Fig. 5 | Thermal conductive model results.** The surface heat flow (**a**) and the temperature gain at depth of 6 km (**b**) and 10 km (**c**) in the thermal conduction model for 150-km thick (blue curves) and 200-km thick (red curves) lithosphere. Solid and dashed lines indicate two different excess temperatures.

**Table 1 | Productive horizons, hydrocarbon maturation and oil/gas resources of the East European platform**

| Basin | Productive horizons | Time of hydrocarbon maturation | Hydrocarbon resources | |
|---|---|---|---|---|
| | | | Oil (10⁹ kg) | Gas (10⁹ m³) |
| Pricaspian | Post-Carboniferous[66] | Permian, occasionally Cretaceous (145–65 Ma) in the south-western basin[47] | 970 (ref. 66) | 1180 (ref. 66) |
| Moscow | Cambrian-Devonian[66] | Unknown[67]; predicted time is Late Cretaceous[68] (100–65 Ma) | Promising oil-gas bearing basins[68,69] | |
| Mezen | Riphean[70] | Unknown[43] | | |
| Timan-Pechora | Devonian-Triassic[66] | Late Permian; occasionally Late Jurassic to Early Cretaceous (160–100 Ma)[43,71] | 4900 (ref. 66) | 2800 (ref. 66) |
| Pre-Uralian | Devonian-Carboniferous[66] | | 14,500 (ref. 66) | 5300 (ref. 66) |
| Volga-Ural | Early Carboniferous[48] | | | |

17 to 40 deg. within 200 My, elevating the temperature of sediments in the EE basins and providing the conditions for hydrocarbon maturation at enhanced heating.

Projecting the estimated extent of the seismically observed PA above the core-mantle boundary[3] toward the lithosphere, an area of at least about $3 \times 10^6$ km$^2$ across the lithosphere would be influenced by the plume's excess heat flux for about 150–200 My. This additional heat source would keep the key organic matter in the oil window during no or insignificant subsidence in the EE basins. The PA plume's presence under Eurasia would also disturb the heat flux in the affected sedimentary basins through underplating and/or hot magma dike injection into the lithosphere promoting again organic matter maturation. Some delay could be expected between the emplacement of the hot material and the time of maturation as the heat diffuses from the hot material up toward the basin.

The rate of hydrocarbon generation is assumed to follow the Arrhenius thermal maturation model[45]. Using this model combined with a hydrological model predicting the temperature and pressure distribution in evolving sedimentary basins, the predicted depth intervals of oil and gas generation are shown[46] to be shallower at lower sedimentation rates because sediments spend a longer time in the same depth interval and are thus more thermally mature. This may explain why at lower sedimentation rates and almost no subsidence in the EE basins since Cretaceous, organic-rich sediments reached higher levels of thermal maturity, generating hydrocarbon accumulations in the EE basins at various depths, including giant oil fields like Tengiz (at depths between 3.8 km and 5.4 km)[47], Kashagan (at a depth of about 4.2 km)[47], and Romashkino (at a depth of about 1.7 km)[48] (see Table 1; Fig. 2). Once formed, the hydrocarbons do not survive if they are submitted to conditions well exceeding those of the oil window. This could happen, for example, if the basin subsidence is rapid and the hydrocarbons progressively attain deeper and therefore hotter levels in the crust. Therefore, the absence of basin subsidence due to the long-term presence of the PA fading plume below Eurasia is an important factor keeping the hydrocarbons basins in the oil window.

In this paper, we have shown that the present PA is likely to be the remnant of a thermally prominent mantle upwelling in the geological past, and that this deep-seated mantle plume influenced the evolution of EE hydrocarbon-bearing basins. As the EE platform moved above the PA plume, vertical tectonic movements recorded in the EE basins show uplifts or insignificant subsidence during the post-Jurassic times. The EE basins have been constantly heated from below for a long geological time (~150–200 My) with elevated temperature and heat flux in the crust. The hydrocarbon basins, including giant oil- and gas-fields, are situated along the tracks of the projection of the PA plume's centre on the surface of the central and eastern parts of the EE platform. Based on modelling of the plume decay, global plate motion models, and a conductive heating model of the lithosphere, we show that the EE basins have been heated above the thermally fading PA plume for the past 150–200 My, creating suitable conditions for hydrocarbon generation.

The study suggests a strong link between a mantle plume heating the thick continental lithosphere for a long geological time, sedimentary basins heated by the weakening plume, and the formation of hydrocarbon provinces in the basins. By the same token, we would expect the generation of oil and gas fields on oceanic passive margins preferentially along the path of hotspots. For example, hydrocarbon resources found in the North Sea and the Norwegian margins could be associated with the Eifel hotspot activity[49] in the Middle-Late Jurassic times resulting in active rifting followed by thermal doming[50]. It may also explain the recent findings of giant resources offshore Brazil and Africa[51] along the track of the Tristan da Cunha plume which begun by the Paraná-Etendeka LIP formation 132 Ma ago and triggered the opening of the South Atlantic Ocean[52]. This suggests that shallow or deep hydrocarbon fields within sedimentary basins are likely to be concentrated in areas of

heat flow enhancements, due in particular to the prolonged presence of a fading mantle upwelling beneath these basins.

## Methods
### Mantle plume modelling
The evolution of mantle plumes originating at the core-mantle boundary is modelled numerically in a bottom heated three-dimensional box $\Omega = [0,3H] \times [0,3H] \times [0,H]$, where $H$ (=2800 km) is the depth of the box. This modelling clarifies the role of thermal diffusion in the evolution of mantle plumes. The equations of momentum conservation (the Stokes equations), mass conservation (the continuity equation), and energy conservation (the heat equation) in an infinite Prandtl number Boussinesq viscous fluid,

$$\nabla P = \nabla \cdot \left[ \eta(T)(\nabla \mathbf{u} + \nabla \mathbf{u}^T) \right] + Ra T \mathbf{e}, \quad \mathbf{x} \in \Omega, \tag{1}$$

$$\nabla \cdot \mathbf{u} = 0, \quad \mathbf{x} \in \Omega, \tag{2}$$

$$\partial T / \partial t + \mathbf{u} \cdot \nabla T = \nabla^2 T, \quad t \in (0,\vartheta), \quad \mathbf{x} \in \Omega, \tag{3}$$

are solved numerically by finite-element and finite-difference methods[21,53]. Here $\mathbf{x}$, $T$, $t$, $\mathbf{u}$, $P$, and $\eta$ are the position vector in 3-D Cartesian coordinates, the temperature, the time, the velocity vector, the pressure, and the viscosity, respectively; $\mathbf{e} = (0,0,1)$ is the unit vector; $Ra = 9.5 \times 10^5$ is the Rayleigh number; symbols $\nabla$ is the gradient operator, $\nabla \cdot$ is the divergence operator, and $^T$ is the transposed matrix. Symmetry and perfect slip conditions are applied to the vertical and horizontal sides of the model box, respectively. The heat flux through the vertical sides is set to zero; the upper and lower sides of the box are assumed to be isothermal surfaces. At the initial time, we assume a depth-dependent temperature stratification in the model. A temperature-dependent viscosity law is employed[54]:

$$\eta(T) = \exp\left( \frac{M}{T + G} - \frac{M}{0.5 + G} \right), \tag{4}$$

where $M = [225/\ln(r)] - 0.25 \ln(r)$, $G = 15/\ln(r) - 0.5$ and $r = 20$ is the ratio between the uppermost and lowermost mantle viscosity.

The model box is divided into $40 \times 40 \times 30$ rectangular finite elements to approximate the velocity potential by tri-cubic splines, and a uniform grid $120 \times 120 \times 90$ is employed for approximation of temperature, velocity, and viscosity. The temperature in the heat equation is approximated by finite differences and determined by the semi-Lagrangian method permitting for relatively large time steps, high accuracy, and low numerical diffusion. A numerical solution to the Stokes equations is based on the introduction of a two-component vector velocity potential and on the application of the Eulerian finite-element method with a tricubic-spline basis for computing the potential. Such a procedure results in a set of linear algebraic equations, which are solved by the conjugate gradient method[53]. The model limitations, uncertainties, and sources of errors are discussed in SD5 of *SI*. The results (isothermal surfaces at 3000 K, i.e. 0.91 of the temperature difference across the mantle) are presented in Fig. 2b–f in a part of the model domain. The hot isotherms illustrate mantle upwelling, while much colder isotherms surrounding the upwelling would indicate downwelling flows (not shown in the figure).

### Global plate motion modelling
The PA position projected on central Eurasia is tracked for 200 My using three models of the GPM[26–28]. Seton et al. (ref. 26) use a combined reference frame: the reference frame[55] for the past 100 My, and the palaeomagnetically-derived true polar wander (TPW) corrected reference frame[56] from 100 Ma to 200 Ma. Müller et al. (ref. 27) follow the model[57] combining a rotation due to two interpolated longitudinal

shift-TPW corrections and an additional correction using the Absolute Plate Motion (APM) rotation poles[56]. Matthews et al. (ref. 28) use also the model[57] and compute an APM between 100 Ma and 250 Ma using the sum between three Euler poles: the first one derived by the palaeo-magnetic poles[58], the second one from a TPW correction[58], and the third one from the same rotation poles as in ref. 27 due to longitudinal shift-TPW correction (see SD3 of SI for detail of the models). The parameters used in the modelling are presented in Supplementary Table S1 of SI.

## Tectonic subsidence analysis

The history of regional crustal tectonic movements in EE sedimentary basins is analysed using 70 exploration boreholes drilled in the Moscow basin, the Timan-Pechora basin, and the Pre-Uralian foredeep (see SD4 of SI). To determine the tectonic subsidence, the backstripping method is employed to separate isostatic effects of sediments and water loading from those of tectonic subsidence[59]. The backstripping procedure consists of stripping stratigraphic units off, starting from the uppermost unit and sequentially removing each underlying unit downwards to the basin's basement. During the procedure, the thickness of the underlying stratigraphic units is corrected for sediment decompaction due to the removal of the load associated with the stripped uppermost unit. Based on the Airy isostasy assessment, the decompacted units are shifted upwards to a prescribed datum accounting for sea-level changes and paleo-water depths. Eustatic sea-level changes were not considered, because no specific paleo-bathymetric data were available for most of the boreholes. Paleo-water depths were predicted based on lithological analysis of sediments and fossils. Hiatuses are treated in most cases as non-deposition events due to uplift. Paleo-bathymetric data used in the backstripping analysis to constrain water-depth estimates through time may influence the resulting subsidence due to uncertainties and errors in the depth determination. While carbonate reefs and erosion surfaces provide rather reliable estimates of paleo-bathymetry, the use of fossil assemblages to estimate paleo-water depths may carry errors[60].

In the backstripping method used here, the isostatic response of the lithosphere is local. This method does not consider the rigidity of the lithosphere and its elastic response to regional loads, which can be calculated using a flexural backstripping method[61]. Namely, for a given sedimentary load, the magnitude of vertical deflection under the load and its lateral wavelength are controlled by the flexural rigidity of the lithosphere (or its effective elastic thickness). Considering the effective elastic thickness of the EE lithosphere to be about 100 km (ref. 62), the deflection of the elastic lithosphere with respect to its hydrostatic deflection is about 30% if the wavelength of the topographic elevation is about 1000 km (ref. 25; see also SD4 of SI). Although the results of the backstripping analysis can be influenced by the method used (e.g., the Airy isostasy vs. the flexural isostasy), we analyse in this paper the tendency of vertical tectonic movements and not their magnitude.

For this aim, we normalise the tectonic subsidence within each individual basin $k$ (=1 for the Moscow basin; 2 for the Timan-Pechora basin; and 3 for the Pre-Uralian foredeep). The normalisation procedure consists of the division of the tectonic subsidence depths $S_i^k(t)$ determined at time $t$ for borehole $i$ (=1, ..., $N_k$; $N_k$ is the number of the wells used in the analysis; $N_1 = 18$, $N_2 = 37$, and $N_3 = 15$) by the maximum tectonic subsidence depth $S_{max}^k$ for the past 210 My in basin $k$, that is, $S_i^k(t)/S_{max}^k < 1$. Tectonic subsidence curves presented in Fig. 4 illustrate the average normalised subsidence at time $t$ [$\hat{S}^k(t) = \sum_{i=1}^{i=N_k}(S_i^k(t)/S_{max}^k)/N_k$]. Vertical lines, crossing these subsidence curves, show the maximum deviation in the normalised subsidence at time $t$ with respect to the average normalised subsidence [$\max_i(S_i^k(t)/S_{max}^k) - \hat{S}^k(t)$].

## Thermal conductive modelling

We consider an analytical model of conductive heat flow in the lithosphere due to its heating from below by a mantle plume. The model temperature is a solution of the 1-D heat conduction equation

$$\partial T/\partial t = \kappa \partial^2 T/\partial z^2 \ (0 < z < h), \tag{5}$$

where $T$ is the temperature, $\kappa$ (=$10^{-6}$ m$^2$ s$^{-1}$) is the thermal diffusion coefficient, $t$ is time, $z$ denotes the depth, and $h$ is the thickness of the lithosphere. The lithosphere's surface is maintained at the same temperature $T_0$ (300 K), and the temperature at the base of the lithosphere is $T_l$. Initially, the lithosphere's temperature has reached steady state and follows the continental geotherm $T_{geoth}(z) = T_0 + 6.55z$, which was modelled using a temperature- and pressure-dependent thermal conductivity and a heat production within the lower crustal rocks of 0.4 μW m$^{-3}$ (ref. 11) to match the surface heat flux 45 mWm$^{-2}$ suitable for the EE basins[63]. At $t = 0$ the temperature at the base of the lithosphere is increased by $\Delta T = T_{p_i} - T_l$ (the plume excess temperature), where $T_{p_i}$ is the plume temperature; two values of the plume temperature are considered, $T_{p_1} = 1770$ K and $T_{p_2} = 1970$ K (ref. 64).

The solution to the heat conduction Eq. (5), satisfying the prescribed initial condition $T_{geoth}$ and boundary conditions $T_0$ at the surface and $T_l$ at the base of the lithosphere, can be analytically derived[65]. The temperature gain in the lithosphere associated with the temperature increase at its base due to mantle plume heating can be expressed as a function of time and depth:

$$T_g(t,z) = \Delta T \left[ \frac{z}{h} + \sum_{n=1}^{\infty} \frac{2 \cos n\pi}{n\pi} \exp\left(-\frac{n^2\pi^2\kappa t}{h^2}\right) \sin\left(\frac{n\pi z}{h}\right) \right]. \tag{6}$$

From this equation we can obtain the surface heat flow gain $Q_s$ as a function of time $t$:

$$Q_s(t) = \frac{k\Delta T}{h} \left[ 1 + \sum_{n=1}^{\infty} 2 \cos n\pi \exp\left(-\frac{n^2\pi^2\kappa t}{h^2}\right) \right], \tag{7}$$

where $k$ (=3 W m$^{-1}$ K$^{-1}$) is the thermal conductivity coefficient. The results discussed below are computed for $n = 30$ (the solution changes insignificantly for larger $n$).

## Data availability

The data supporting the conclusions of this article (GPM-based PA tracks, the geographic location of drilled boreholes, original and normalised tectonic subsidence data, modelled temperature gain and surface heat flow data) are provided as source data files and can be downloaded from the website: https://github.com/aizadeh/EE_PA_dataset.git.

## Code availability

The code to compute a numerical model of mantle plume diffusion and a thermal conduction model can be made available upon request from the authors.

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

## Acknowledgements
Helpful discussion with V. Courtillot, Y. Galushkin, M. Greff-Lefftz, J.-L. Le Mouël, V. Pavlov, and G. Schubert and constructive comments by four reviewers enable us to improve the manuscript. I. Tsepelev is thanked for his assistance in numerical modelling of the plume evolution. I. Artemieva provided the LAB data on the EE platform. A.I.Z. discloses a support for publication of this work from the Helmholtz Association (as a part of MTET-Geoenergy program).

## Author contributions
A.I.-Z. and A.D. conceptualise the paper. A.I.-Z. contributed to the development of numerical geodynamic and analytical thermal models, analysed the data related to tectonic subsidence of sedimentary basins, and performed related modelling. A.D. contributed to the combined analysis of seismic-tomographic and geodynamic data, and to the analysis of thermal models. J.B. contributed to the development and analysis of the tracks of the Perm Anomaly upwelling projections on the surface of western Eurasia. Y.V. contributed to analysis of the evolution of sedimentary basins, related magmatism, and hydrocarbon maturation. All authors contributed to the interpretation of the results. A.I.-Z. performed writing – original draft preparation and its revision; A.I.-Z. and A.D. prepared the figures. All co-authors performed writing – review and editing – and contributed to the revision of the initial manuscript.

## Funding

## Competing interests
The authors declare no competing interests.
