## [Peer Review File · Nature Communications]

REVIEWER COMMENTS

Reviewer #1 (Remarks to the Author):

Review of “East European Sedimentary Basins Stewed Over an Ancient Mantle Upwelling” by Ismail-Zadeh et al.

The manuscript presents an interesting idea, which I enjoyed reading. The evolution of a mega-plume, now extinct, can explain sedimentary basin features not directly available in lithospheric evolutionary models. It is well known that the large basins overlying the East European craton contains thick sedimentation and long-lived unconformities that cannot be directly explained by mechanical models of lithospheric evolution. The little amount of fault-driven basin accommodation space observed to be created over geological time in many EE basins cannot justify the higher temperatures measured in the organic maturation of sediments. An additional gain of temperature over the observed mechanical subsidence and sediment burial is required in the structure of the basins to explain thermal markers such as the organic maturation. The wavelength of these basins may be considered large enough to be regionally influenced by a mega-plume. Because I believe that sub-lithospheric forcing is the way to research the observed features, I recommend publication of the novel idea presented in the manuscript. However, the idea is speculative given available information, and the authors should be less bold in their conclusions, by acknowledging better error bars and potential pitfalls. ISuch changes can be performed in a minor revision.

Suggestions:

a) All assumptions and modelling performed in the manuscript are based on a viscous structure. Sure that this is an appropriate approach for the sub-lithospheric mantle, but this is not appropriate for the EE lithosphere. The EE craton has presently one of the thickest elastic structures. This translates at least in two approaches of the manuscript:

- Although not well explained in the manuscript (kind of a black box in the text and appendix), the subsidence analysis appears to be simple isostatic backstripping. This is obviously inappropriate give the high elastic thicknesses of the EE craton (often more than 80 km) and flexural isostatic backstripping must be assessed because errors are certainly higher than 50%. I do not understand very well Fig. 4a, it may be that the authors refer only to unconformities in the basins. I suggest explaining better the subsidence analysis and estimate the errors bars induced by flexure (both in subsidence analysis and in other potential far-field induced effects), while providing further explanations or redrawing Fig. 4a;

- At the higher values or enrichment of 100 degrees in the basins and given the high elasto-plastic composition, one would expect to see much larger active rifting (in the Ziegler sense, plume-impinged rifting) than the minor one observed in the basins. Ideally one should perform more an elasto-visco-plastic above a mantle plume head type of modelling, but I suggest that the authors discuss at least the influence of the EE craton elasticity in the dynamic lithospheric evolution.

b) I suggest acknowledging and explaining better the resolution and error bars of the assumptions, which is the wavelength of the inferred plume during its evolution. Anything higher resolution is possibly inappropriate in terms of interpretation.

c) While the idea is certainly applicable to the EE craton basins, with the methods and tools of this

manuscript the speculative inferences at the end of the manuscript to regions undergoing large scale kinematic movements, such as the North Sea, Norwegian margin or offshore Atlantic are certainly too speculative and inappropriate. Plumes are not the only mechanism able to explain over-maturation of sediments. In such regions, alternative explanations (such as the asymmetry of the extension or exhumation during the formation of hyper-extended margins) are already proven to be valid scenarios. Any plume contribution must be first demonstrated at higher resolution than what the manuscript provides. I suggest removing these speculative considerations, which in my view decrease the quality of the manuscript.

d) Far-field stresses or lithospheric folding have been used in literature to explain regional EE craton basins geometry. One should shortly include these ideas in the discussion.

Other smaller suggestions:

- Would be good to be specific which Tethys you are speaking about, because not all has been active in the last 300Ma. It is true that between Japan and Africa the only true large scale noteworthy features are the Pacific and Aegean subduction system, but the latter is certainly not 300 Ma old.
- Although fancy, Fig. 1 is not very readable in the sense of understanding the structure. I do not have a good alternative solution, but maybe a depth slice with a cross-section would be more helpful?
- The thermal model has also little explanation, although references are partly available. If there is space, I suggest considering explaining with more detail the numerical implementation.
- Please explain more to the point the link between the mechanism proposed and the formation of mega-petroleum provinces. I guess the assumption is that because of the large wavelength, the mechanisms affect larger basinal areas, creating mega-plays. Right?
- For the reasons explained above, the influence of the plume appears to me overstated. It is better to say simply that the additional heat source “nudge” the temperature of key organic matter accumulation in the hydrocarbon window.
- Second page, second paragraph, you are either specific or remove “some other geophysical processes.”
- Fig. 3a – Is it possible to change the grey area in Fig 3a with more specific evolutionary colour zones? Such as one key moment in the plume spatial evolution?

Reviewer #2 (Remarks to the Author):

This paper presents the intriguing hypothesis that the large hydrocarbon fields of the East European basin may have been formed owing to the heat provided by the remnants of the head of a plume whose root at the core-mantle boundary would presently be manifested by the tomographically imaged low shear velocity “PERM” anomaly.

In particular, the model explains the lack of sedimentary basin subsidence that is generally invoked in models of hydrocarbon reservoir formation.

If this interpretation is correct, it should be good news to those working on imaging deep mantle plumes, as it would add to the relevance of their work relevant to societally important near surface geology problems.

The geodynamic plume modeling showing how a plume can become separated from its root, and the thermodynamic modeling are sound and interesting.

What is missing in this paper is a broader picture of the relationship through time of the “PERM” plume with the Eurasian lithosphere. Indeed, before getting broken up into a separate tail and head, the plume must have been continuous and upon reaching the base of the lithosphere, it must have interacted with it for a long time. The authors cite two rather different hypotheses for links between the present day PERM anomaly and near surface observations of ancient volcanism: Siberian Traps, versus the more southerly Emeishan Volcanics. These two hypotheses rely on very different “views” on the evolution of mantle plumes. The former has a plume fixed in the deep mantle since at least Permian times, the latter has the plume moving around. This deserves further discussion and clarification: the statement “The longevity of these downwellings and mass conservation in the mantle imply the longevity of the hot upwelling” is incorrect at the level of a single plume. The upwelling part of the flow could have changed location and therefore also the place of impact at the base of the lithosphere, unless the model considered is one of a plume anchored at the core-mantle boundary by a dense root (e.g. Jellinek and Manga, 2005). It is therefore not clear what the general framework of the reasoning is in this paper. Is the plume fixed in its geographical position and fading with time in its strength, or is it moving around as well as fading? Is it more likely related to the Siberian Traps or the Emeishan Volcanics? What makes the studied area special for the manifestation of the plume head activity? A geological map showing how the EE platform might have aligned through time with the plume head, or some arguments as to why the plume head would have moved and settled beneath the EE platform could help clarify this. Figure 2 is not very helpful, nor is Figure 3, which is not showing the geology.

More generally, the figures are awful. There must be a better way to represent the relation of the deep mantle structure to the lithosphere than what is shown in Fig 1 which is hardly understandable. Figure 3b implies that the plume head has moved around geographically, when, unless I missed the point, what the authors want to show is the absolute plate motion of the Eurasian plate with respect to the present day location of the PERM anomaly. Figure 4, which shows a welcome sketch of the different basins in the EE platform (But no other geological details) is plotted in a different projection than Fig 3a and the location of the PERM anomaly is not even indicated.

In summary, the model of heating due to a plume to explain the lack of subsidence (this one is clearly presented) of the EE basins in the last 150 My is plausible, but the link to the PERM anomaly, while attractive and plausible, should be further clarified.

Reviewer #3 (Remarks to the Author):

This thought-provoking article proposes that a plume associated with the Perm Anomaly could explain the evolution of the East European platform from about 250 million years ago. My understanding of the proposed scenario is that a mantle plume rising from the Perm Anomaly would have been strongest when it first reached the shallow mantle about 250 million years ago (Permo-Triassic), possibly causing the Siberian Traps. Laboratory and numerical experiments show that the plume could have weakened

over time. Plate motions deduced from global tectonic reconstructions show that the East European platform could have been influenced by a weakening plume above the stationary Perm Anomaly from 200 million years ago. The analysis of borehole data (backstripping) reveals a lack of tectonic subsidence of basins from the East European platform from 150 million years ago, and this is attributed to dynamic support from a weakening mantle plume in the absence of volcanism due to the thickness of the East European platform lithosphere. Thermal modelling shows that a thermally weakening plume beneath the East European platform could have led to conditions favourable for hydrocarbon maturation.

I have several questions and comments:

- the region of interest is roughly centred on 55°E/55°N. The Siberian Traps are roughly centred on 85°E/65°N. The distance between these two points is about 2,000 km. How could a mantle plume linked to the Perm Anomaly have caused the Siberian Traps around 250 million years ago and influenced the East European platform from about 250 million years ago (Figure 2a of the manuscript)?

- the Kola-Dnieper large igneous province (e.g. Ernst et al., 2020) erupted at ~370-360 Ma and is centred on 57°E/40°N. The distance between the two points is only 925 km, and the Kola-Dnieper large igneous province is in the longitudinal vicinity of the East European platform.

- volcanism (which is part of the Kola-Dnieper large igneous province) is documented during the period of rifting of the basins of the East European platform around 370 million years ago (see Wilson and Lyashkevich, 1996 and supplementary material for the subsidence)

- based on the above, could an earlier mantle plume (about 120 million years earlier than the proposed mantle plume) have been associated with volcanism, rifting and subsidence in the East European platform?

- is the lack of subsidence of the East European platform from about 160 million years ago uniquely explained by dynamic support from a weakening mantle plume?

- it might be worth mentioning that the present-day residual topography and dynamic topography of the East European platform are negative, and explaining how this fits with the proposed scenario.

- the motion of plates is considered (Figure 3), however, the Perm Anomaly is assumed to have been stationary from 200 million years ago. Is this a reasonable assumption?

References:

Ernst, R.E., Rodygin, S.A. and Grinev, O.M., 2020. Age correlation of Large Igneous Provinces with Devonian biotic crises. *Global and Planetary Change*, 185, p.103097.

Wilson, M. and Lyashkevich, Z.M., 1996. Magmatism and the geodynamics of rifting of the Pripyat-Dnieper-Donets rift, East European Platform. *Tectonophysics*, 268(1-4), pp.65-81.

“East European sedimentary basins stewed over an ancient mantle upwelling”

Response to the reviewers' comments

The authors are very grateful to three reviewers and to the Editor for thorough review of the initial manuscript of the paper and very constructive comments. Below is a point-to-point response (marked by red below) to the reviewers' comments. We refer to lines (ll.) of the revised manuscript and to the revised Supplementary Information, where the revision is marked by red.

Reviewer #1 (Remarks to the Author):

The manuscript presents an interesting idea, which I enjoyed reading. The evolution of a mega-plume, now extinct, can explain sedimentary basin features not directly available in lithospheric evolutionary models. It is well known that the large basins overlying the East European craton contains thick sedimentation and long-lived unconformities that cannot be directly explained by mechanical models of lithospheric evolution. The little amount of fault-driven basin accommodation space observed to be created over geological time in many EE basins cannot justify the higher temperatures measured in the organic maturation of sediments. An additional gain of temperature over the observed mechanical subsidence and sediment burial is required in the structure of the basins to explain thermal markers such as the organic maturation. The wavelength of these basins may be considered large enough to be regionally influenced by a mega-plume. Because I believe that sub-lithospheric forcing is the way to research the observed features, I recommend publication of the novel idea presented in the manuscript. However, the idea is speculative given available information, and the authors should be less bold in their conclusions, by acknowledging better error bars and potential pitfalls. ISuch changes can be performed in a minor revision.

Suggestions:

a) All assumptions and modelling performed in the manuscript are based on a viscous structure. Sure that this is an appropriate approach for the sub-lithospheric mantle, but this is not appropriate for the EE lithosphere. The EE craton has presently one of the thickest elastic structures.

This translates at least in two approaches of the manuscript:

- Although not well explained in the manuscript (kind of a black box in the text and appendix), the subsidence analysis appears to be simple isostatic backstripping.

A description of the backstripping procedure used in this work has been added to the section *Methods* (lines 366-379).

This is obviously inappropriate give the high elastic thicknesses of the EE craton (often more than 80 km) and flexural isostatic backstripping must be assessed because errors are certainly higher than 50%.

The influence of flexural rigidity has been now estimated and discussed in *Methods* (ll. 380-390), and in more detail in *Supplementary Discussion 3* (ll. S146-S164).

I do not understand very well Fig. 4a, it may be that the authors refer only to unconformities in the basins. I suggest explaining better the subsidence analysis and estimate the errors bars induced by flexure (both in subsidence analysis and in other potential far-field induced effects), while providing further explanations or redrawing Fig. 4a

The normalization procedure applied to the subsidence curves and the maximum variations in the vertical tectonic movements are now discussed in *Methods* (ll. 391-399).

- At the higher values or enrichment of 100 degrees in the basins and given the high elasto-plastic composition, one would expect to see much larger active rifting (in the Ziegler sense, plume-impinged rifting) than the minor one observed in the basins. Ideally one should perform more an elasto-visco-plastic above a mantle plume head type of modelling, but I suggest that the authors discuss at least the influence of the EE craton elasticity in the dynamic lithospheric evolution.

In our analytical thermal models, the thermal enrichment ranges from ~20 to ~100 degrees. Considering the excess temperature to decrease with geological times over 200 My due to the plume's thermal decay, the enrichment will be not enough to generate plume-impinged rifting of the EE lithosphere. A discussion is added (ll. 232-234)

b) I suggest acknowledging and explaining better the resolution and error bars of the assumptions, which is the wavelength of the inferred plume during its evolution. Anything higher resolution is possibly inappropriate in terms of interpretation.

Uncertainties, errors, and limitations in the wavelength of the PA plume (ll. 143-147) as well as in tectonic subsidence analysis (ll. 375-379) and in the modeling of mantle plume evolution (see *Supplementary Discussion 4*) are now discussed in more detail.

c) While the idea is certainly applicable to the EE craton basins, with the methods and tools of this manuscript the speculative inferences at the end of the manuscript to regions undergoing large scale kinematic movements, such as the North Sea, Norwegian margin or offshore Atlantic are certainly too speculative and inappropriate. Plumes are not the only mechanism able to explain over-maturation of sediments. In such regions, alternative explanations (such as the asymmetry of the extension or exhumation during the formation of hyper-extended margins) are already proven to be valid scenarios. Any plume contribution must be first demonstrated at higher resolution than what the manuscript provides. I suggest removing these speculative considerations, which in my view decrease the quality of the manuscript.

As we have no strong evidence for influence of mantle plume on the hydrocarbon maturation in the North Sea, Norwegian margin or offshore Atlantic regions, the relevant sentence is removed from the manuscript.

d) Far-field stresses or lithospheric folding have been used in literature to explain regional EE craton basins geometry. One should shortly include these ideas in the discussion.

We have included a discussion on the mechanisms of EE basin evolution in *Supplementary Discussion 3* (ll. S130-S139).

Other smaller suggestions:

- Would be good to be specific which Tethys you are speaking about, because not all has been active in the last 300Ma. It is true that between Japan and Africa the only true large scale noteworthy features are the Pacific and Aegean subduction system, but the latter is certainly not 300 Ma old.

“Tethys” was replaced by “Neo-Tethys”.

- Although fancy, Fig. 1 is not very readable in the sense of understanding the structure. I do not have a good alternative solution, but maybe a depth slice with a cross-section would be more helpful?

Figure 1 is revised and includes now cross sections.

- The thermal model has also little explanation, although references are partly available. If there is space, I suggest considering explaining with more detail the numerical implementation.

We have added a description of the numerical implementation in *Methods* (ll. 338-345).

- Please explain more to the point the link between the mechanism proposed and the formation of mega-petroleum provinces. I guess the assumption is that because of the large wavelength, the mechanisms affect larger basinal areas, creating mega-plays. Right? For the reasons explained above, the influence of the plume appears to me overstated. It is better to say simply that the additional heat source “nudge” the temperature of key organic matter accumulation in the hydrocarbon window.

The PA mantle plume activity has influenced a wide area (about 3×10^6 km²) of the EE lithosphere above the PA. And we argue that the excess of the heat flux due to the presence of the plume beneath the region for about 150-200 My provided the additional heat source to keep organic matter in the oil window. A clarification is inserted (ll. 265-269; 281-283).

- Second page, second paragraph, you are either specific or remove “some other geophysical processes.”

Removed

- Fig. 3a – Is it possible to change the grey area in Fig 3a with more specific evolutionary colour zones? Such as one key moment in the plume spatial evolution?

Figure 3 is revised and includes now the tracks showing projections of the plume onto the surface at different times.

Reviewer #2 (Remarks to the Author):

This paper presents the intriguing hypothesis that the large hydrocarbon fields of the East European basin may have been formed owing to the heat provided by the remnants of the head of a plume whose root at the core-mantle boundary would presently be manifested by the tomographically imaged low shear velocity “PERM” anomaly.

In particular, the model explains the lack of sedimentary basin subsidence that is generally invoked in models of hydrocarbon reservoir formation.

If this interpretation is correct, it should be good news to those working on imaging deep mantle plumes, as it would add to the relevance of their work relevant to societally important near surface geology problems.

The geodynamic plume modeling showing how a plume can become separated from its root, and the thermodynamic modeling are sound and interesting.

What is missing in this paper is a broader picture of the relationship through time of the “PERM” plume with the Eurasian lithosphere. Indeed, before getting broken up into a separate tail and head,

the plume must have been continuous and upon reaching the base of the lithosphere, it must have interacted with it for a long time.

The numerical modeling of thermal plume evolution presents the snapshots of the Permian-Triassic (revised Fig. 2a) to the Cenozoic (revised Fig. 2e) times. This model assumes that the plume was active some 250 My ago and was fading thermally since Early Triassic. We do not link the plume to any specific area on the surface of the lithosphere in the geological past before 200 My ago, because uncertainties in paleolatitudes may pollute the GPM reconstructions beyond this geological time frame (Bess & Courtillot, 1991). It is discussed in *Supplementary Discussion 1* (ll. S24-S28)

The authors cite two rather different hypotheses for links between the present day PERM anomaly and near surface observations of ancient volcanism: Siberian Traps, versus the more southerly Emeishan Volcanics. These two hypotheses rely on very different “views” on the evolution of mantle plumes. The former has a plume fixed in the deep mantle since at least Permian times, the latter has the plume moving around. This deserves further discussion and clarification: the statement “The longevity of these downwellings and mass conservation in the mantle imply the longevity of the hot upwelling” is incorrect at the level of a single plume. The upwelling part of the flow could have changed location and therefore also the place of impact at the base of the lithosphere, unless the model considered is one of a plume anchored at the core-mantle boundary by a dense root (e.g. Jellinek and Manga, 2005). It is therefore not clear what the general framework of the reasoning is in this paper. Is the plume fixed in its geographical position and fading with time in its strength, or is it moving around as well as fading? Is it more likely related to the Siberian Traps or the Emeishan Volcanics? What makes the studied area special for the manifestation of the plume head activity?

As the topic of the paper is not related to the origin of the Perm Anomaly, we would not like to debate on the origin in this article, but some comments on this topic are inserted in *Supplementary Discussion 1* (ll. S16-S69). Also, as we assume that a mantle plume associated with the PA is fixed in space, the reference to the Emeishan Volcanics is removed from the main text and added to *Supplementary Discussion 1* (ll. 39-43).

A geological map showing how the EE platform might have aligned through time with the plume head, or some arguments as to why the plume head would have moved and settled beneath the EE platform could help clarify this.

Figure 2 is not very helpful, nor is Figure 3, which is not showing the geology.

Paleogeographic maps are inserted into *Supplementary Information* (see *Supplementary Fig. 1*), which also presents the geological features of the EE platform since the Late Permian times.

More generally, the figures are awful. There must be a better way to represent the relation of the deep mantle structure to the lithosphere than what is shown in Fig 1 which is hardly understandable.

Figure 1 is redrawn.

Figure 3b implies that the plume head has moved around geographically, when, unless I missed the point, what the authors want to show is the absolute plate motion of the Eurasian plate with respect to the present day location of the PERM anomaly.

Figure 3 is revised, and illustrates the traces of the piercing point on the moving Eurasian lithosphere of a fixed plume associated with the PA.

Figure 4, which shows a welcome sketch of the different basins in the EE platform (But no other geological details) is plotted in a different projection than Fig 3a and the location of the PERM anomaly is not even indicated.

Figure 4 is revised.

In summary, the model of heating due to a plume to explain the lack of subsidence (this one is clearly presented) of the EE basins in the last 150 My is plausible, but the link to the PERM anomaly, while attractive and plausible, should be further clarified.

We have clarified the link in *Supplementary Discussion 1*.

Reviewer #3 (Remarks to the Author):

This thought-provoking article proposes that a plume associated with the Perm Anomaly could explain the evolution of the East European platform from about 250 million years ago. My understanding of the proposed scenario is that a mantle plume rising from the Perm Anomaly would have been strongest when it first reached the shallow mantle about 250 million years ago (Permo-Triassic), possibly causing the Siberian Traps. Laboratory and numerical experiments show that the plume could have weakened over time. Plate motions deduced from global tectonic reconstructions show that the East European platform could have been influenced by a weakening plume above the stationary Perm Anomaly from 200 million years ago. The analysis of borehole data (backstripping) reveals a lack of tectonic subsidence of basins from the East European platform from 150 million years ago, and this is attributed to dynamic support from a weakening mantle plume in the absence of volcanism due to the thickness of the East European platform lithosphere. Thermal modelling shows that a thermally weakening plume beneath the East European platform could have led to conditions favourable for hydrocarbon maturation.

I have several questions and comments:

- the region of interest is roughly centred on 55°E/55°N. The Siberian Traps are roughly centred on 85°E/65°N. The distance between these two points is about 2,000 km. How could a mantle plume linked to the Perm Anomaly have caused the Siberian Traps around 250 million years ago and influenced the East European platform from about 250 million years ago (Figure 2a of the manuscript)?

This paper presents the evolution of the mantle plume associated with the Perm Anomaly (PA) from 200 My ago until present. We assume that the plume was active and prominent before it, and while we do not deal with the PA origin, we consider that this plume could cause the Siberian traps at the Permian-Triassic boundary (e.g., Torsvik et al., 2016; also see *Supplementary Discussion 1*). As discussed in the manuscript, the East European (EE) lithosphere moved northeastward at the rate of about (5N, 10E) in My in the Middle to Late Triassic according to paleogeographic/palinspastic (Kazmin & Natapov, 1998) reconstructions (see *Supplementary Fig. 1*). Hence, the EE lithosphere reached the head of this mantle plume by the Early Jurassic. We should mention that the evolution of the plume since the Early Jurassic is of our research interest.

- the Kola-Dnieper large igneous province (e.g. Ernst et al., 2020) erupted at ~370-360 Ma and is centred on 57°E/40°N. The distance between the two points is only 925 km, and the Kola-Dnieper large igneous province is in the longitudinal vicinity of the East European platform.

According to the paleogeographic/paleomagnetic reconstructions by the Kuzmin et al. (2010) (see also *Supplementary Fig. 2* and relevant *Supplementary Discussion 1*, ll. S43-S48), Kola-Dnieper LIP is associated with the TUZO LLSVP, its eruption site 360 My ago was in the vicinity of the equator, and hence KDLIP is not related to the present Perm Anomaly. According to Ernst (2014), active volcanism in the region spans from 380 to 360 My. It means that this plume could start fading thermally soon after its active volcanic phase, its thermal influence on the lithosphere would be decaying with time, and hence, even though it would be located in the vicinity of the PA, its impact on the EE lithosphere would be insignificant (if any) because of thermal diffusion of the plume.

- volcanism (which is part of the Kola-Dnieper large igneous province) is documented during the period of rifting of the basins of the East European platform around 370 million years ago (see Wilson and Lyashkevich, 1996 and supplementary material for the subsidence).

The reference to the earlier magmatic events in the region is added to *Supplementary Discussion 3* (ll. S193-S199).

- based on the above, could an earlier mantle plume (about 120 million years earlier than the proposed mantle plume) have been associated with volcanism, rifting and subsidence in the East European platform?

An earlier mantle plume may influence the evolution of the EE platform, and the Kola-Dnieper mantle plume associated with the KDLIP is a bright example of the impact on the EE platform in the Late Devonian times (see *Supplementary Discussion 3*, ll. S193-S199 and S235-S239). Meanwhile, we study the evolution of the region since the Jurassic and the Kola-Dnieper plume have no influence on the EE platform during this time.

- is the lack of subsidence of the East European platform from about 160 million years ago uniquely explained by dynamic support from a weakening mantle plume?

Our scientific hypothesis is supported by observations and model results, and that is what we present in this paper. Meanwhile, an alternative hypothesis may appear in the future with more data and more new knowledge on the region, which could also explain widespread unconformities in the EE basins for about two hundred My.

- it might be worth mentioning that the present-day residual topography and dynamic topography of the East European platform are negative, and explaining how this fits with the proposed scenario.

An acceleration in subsidence during the Quaternary times is observed in some borehole data. There is no direct link between this subsidence and the proposed evolution of the region. The subsidence occurred during a short geological time to understand the mechanisms of this subsidence. Also, some discussion on tectonic subsidence of the EE basins since Devonian is added in *Supplementary Discussion 3* (ll. S165-S175).

- the motion of plates is considered (Figure 3), however, the Perm Anomaly is assumed to have been stationary from 200 million years ago. Is this a reasonable assumption?

Yes. Torsvik et al. (2016) provided the evidence for this on the basis of the stability of large low-shear-velocity provinces. This is discussed in *Supplementary Discussion 1* (ll. S39-S43).

Reviewers' comments:

Reviewer #1 (Remarks to the Author):

The authors have responded adequately and have considered correctly all my comments. Therefore I suggest manuscript acceptance at this time.

Reviewer #2 (Remarks to the Author):

This is a much improved version of the original manuscript. I only have two comments of significance:

1) While figure 1 is much clearer, it is still a bit confusing because each panel is showing a map at 2800 km projected with a different center and two different models depending on the panel. I suggest showing the same cross section in the two models.

2) Because this paper is still to a large extent speculative, I would tone down the assertions in the abstract:

- a. "we show here that the EE lithosphere HAS BEEN situated..." replace "has been" by "MAY HAVE BEEN"
- b. "This establishes a profound relationship" : replace "establishes" by "suggests".

Finally, the paper would benefit from being read by a native English speaker. It is not critical, but would help with smooth reading (in particular making sure the tense of the verbs on lines 10-167 are consistent: gains-> gained, line 159: starts->started

Line 173: "shrinks"-> shrunk etc.. and in quite a few places the ordering of words in the sentence, as well as missing or superfluous articles.

Reviewer #3 (Remarks to the Author):

I am not convinced by the geological and geophysical evidence presented in this manuscript. The proposed scenario is that a mantle plume existed above the fixed Perm Anomaly, triggered the Siberian Traps 250 million years ago, and then affected the subsidence and thermal evolution of the East European platform from 200 million years ago.

Problems with this scenario include (i) that no plume is imaged above the Perm Anomaly at present-day, (ii) that there is no volcanism in the East European platform, (iii) that the link between the Perm Anomaly and the Siberian Traps is not demonstrated in the manuscript, and (iv) that deep basal mantle structures may have been mobile over time.

The scenario presented in Figure 2 suggests a solution to problem (i) in which the lifetime of the plume tail would be limited to 50 million years or less (specifically, Figure 2a,b,c). If this is the case, why are so many plume tails long-lived as demonstrated by oceanic hotspot tracks that span about 100 million years (see for example O'Neill et al., 2005)? As opposed to plume tails, plume heads are thought to be short-lived and to result in the eruption of large igneous provinces over 1 to 5 million years (Bryan and Ernst,

2008). In contrast, in Figure 2, the plume head persists for at least 200 million years. Is the model presented in Figure 2 applicable to Earth?

Regarding problem (ii), it is argued that the plume was waning and that the East European lithosphere is thick, preventing melts from reaching the surface (l. 131-134). Davies et al. (2015) documented a long hotspot track beneath the eastern part of the Australian continent, and they showed that melting occurred where the lithosphere was less than 150 km thick. It is unclear that the thickness of the Eastern European lithosphere is more than 150 km in the eastern part of the region of interest (see for example Steinberger and Becker, 2018 or the global lithospheric thickness models available from Hoggard et al., 2020 at <https://osf.io/twksd/>).

The link between the Perm Anomaly and the Siberian Traps is proposed based on a reconstruction from 1998 (Supplementary Figure 1) based on which it is argued that 'the EE platform moved north-eastward at the rate of about (5°N, 10°E) in My in the Middle to Late Triassic times.' (l. 83-84). It would be helpful to quote a rate in degrees per million years. It would also be helpful to link the Perm Anomaly to the Siberian Traps as in Figure 3 of the main text (which links the Perm Anomaly to the Eastern European platform) by using a tectonic reconstruction that extends at least back to 250 million years ago (examples include the reconstructions of Matthews et al., 2016 or Young et al., 2019). In the rebuttal, it is argued that the focus of the study is the past 200 million years, however, in the absence of geophysical evidence for a plume and of any volcanism linked to the Perm Anomaly in the last 250 million years, it seems reasonable to request some geological evidence linking the Perm Anomaly to a mantle plume. If the Perm Anomaly is linked to the Siberian Traps, should there be some volcanism between the Siberian Traps and the East European platform in places where the lithosphere is less than 150 km thick?

While it would be easier to link deep seismically slow anomalies to past volcanic eruptions if the structure of the deep Earth was rigid and fixed over time, Flament et al. (2022) showed that volcanic eruptions could be linked to either fixed or mobile deep mantle structures.

References

Bryan, Scott E., and Richard E. Ernst. "Revised definition of large igneous provinces (LIPs)." *Earth-Science Reviews* 86.1-4 (2008): 175-202.

Davies, D. R., et al. "Lithospheric controls on magma composition along Earth's longest continental hotspot track." *Nature* 525.7570 (2015): 511-514.

Flament, Nicolas, et al. "Assembly of the basal mantle structure beneath Africa." *Nature* 603.7903 (2022): 846-851.

Hoggard, Mark J., et al. "Global distribution of sediment-hosted metals controlled by craton edge stability." *Nature Geoscience* 13.7 (2020): 504-510.

O'Neill, Craig, Dietmar Müller, and Bernhard Steinberger. "On the uncertainties in hot spot reconstructions and the significance of moving hot spot reference frames." *Geochemistry, Geophysics*,

Geosystems 6.4 (2005).

Steinberger, Bernhard, and Thorsten W. Becker. "A comparison of lithospheric thickness models." *Tectonophysics* 746 (2018): 325-338.

Young, Alexander, et al. "Global kinematics of tectonic plates and subduction zones since the late Paleozoic Era." *Geoscience Frontiers* 10.3 (2019): 989-1013.

Response to the Reviewers comments

The authors are grateful to the Reviewers for their comments. We provide a point-by-point response to their comments below. Their comments are marked by red.

Reviewer #1:

The authors have responded adequately and have considered correctly all my comments. Therefore I suggest manuscript acceptance at this time.

The authors are grateful to Reviewer #1 for the recommendation on acceptance.

Reviewer #2:

This is a much improved version of the original manuscript. I only have two comments of significance:

1) While figure 1 is much clearer, it is still a bit confusing because each panel is showing a map at 2800 km projected with a different center and two different models depending on the panel. I suggest showing the same cross section in the two models.

Done. Figure 1 has been revised as suggested by the reviewer.

2) Because this paper is still to a large extent speculative, I would tone down the assertions in the abstract:

a. “we show here that the EE lithosphere HAS BEEN situated...” replace “has been” by “MAY HAVE BEEN”

b. “This establishes a profound relationship” : replace “establishes” by “suggests”.

We have revised the wording in *a* and *b*, respectively.

A certain level of speculation exists in many published works, and our work is based on a hypothesis as well. Namely, we propose that a present seismic observation (the Perm Anomaly, PA) at the base of the mantle is linked to geological observations at the surface. Geological interpretations of seismic tomography images are challenging because of the limitation of our knowledge regarding the geodynamic evolution of the region and its surroundings, and some uncertainties associated with the inversion/assimilation of data both in seismology and geodynamics (e.g., Ismail-Zadeh et al., 2023). Nevertheless, the present seismic evidence indicates that the PA is separated from the other two large provinces of low seismic velocities (below the Pacific and the Indo-Atlantic regions) by the Neo-Tethys subduction to the south and the Pacific subduction to the north and the east. Geology and paleomagnetism show that the Neo-Tethys and Pacific subductions are long-lived features that have been active at least for the last 300 My, delivering a nearly continuous downwelling flow of cold material on the core-mantle boundary around the PA slow seismic velocity area, and therefore delimiting a third “box” in the mantle. The longevity of these downwellings and mass conservation in the mantle imply a nearly continuous generation of hot upwellings within the PA box, as seen in our numerical simulations and laboratory experiments.

Based on the hypothesis that the PA is an ancient mantle structure linked to an old mantle plume, we have studied the thermal decay of this old plume using numerical modelling and laboratory experiments, as the first step, in supporting our findings, which are: (1) the East European (EE) lithosphere has been situated over the weakening PA upwelling for about 150-200 million years (this was obtained by careful paleomagnetic, paleogeographic and global plate motion reconstructions). (2) As the EE platform moved above PA in post-Jurassic times, the vertical tectonic movements recorded in sedimentary hydrocarbon-rich basins show either hiatus/uplift or insignificant subsidence (based on the tectonic analysis of EE sedimentary basins). (3) The basins have been thermally stewing above the PA upwelling, creating suitable conditions for hydrocarbon maturation (based on analytical modelling of heat conduction through the lithosphere and conditions for hydrocarbon maturation). We have presented in the manuscript convincing scientific evidence for each of the three findings, which supports our conclusions. Namely, based on the plume decay model, global plate motion models, and conductive heating model inside the lithosphere, we have shown that the East European basins have been stewing thermally above the weakening PA plume for the last 150-200 My, creating suitable conditions for hydrocarbon generation.

One can still speculate about the possible limitations and uncertainties associated with our finding, for example, (i) whether the PA is a relatively recent negative seismic wave anomaly or it is a remnant feature of a mantle plume; (ii) if the PA is a fading mantle plume feature, whether it was linked to the Siberian trap

area or to other plumes in the past; and (iii) whether a thermally fading mantle plume can generate magmatic events, especially in the case of the thick lithosphere. Karl Popper in his work “Alles Leben ist Problemlösen” (1994) said: “Although we do our best in science to find out the truth, we are conscious of the fact that we can never be sure to have found the truth. We know that our scientific theories always remain hypotheses, but that in many cases, we can find out whether or not a new hypothesis is superior to an old one.” Therefore, only further investigations can confirm our findings or reject them based on new evidence.

Finally, the paper would benefit from being read by a native English speaker. It is not critical, but would help with smooth reading (in particular making sure the tense of the verbs on lines 10-167 are consistent: gains->gained, line 159: starts->started

Line 173: “shrinks”-> shrunk etc.. and in quite a few places the ordering of words in the sentence, as well as missing or superfluous articles.

We have been careful to revise the sentences accordingly. Also, we made further stylistic, orthographic, and syntactic revision of the text. And if the manuscript is accepted and the Editor recommends further English improvement, the authors would take care of it.

Reviewer #3:

I am not convinced by the geological and geophysical evidence presented in this manuscript. The proposed scenario is that a mantle plume existed above the fixed Perm Anomaly, triggered the Siberian Traps 250 million years ago, and then affected the subsidence and thermal evolution of the East European platform from 200 million years ago.

We are very sorry that we could not convince you. Let us start with the scenario you have mentioned above. First at all, we did not propose this scenario. The hypothesis that the Perm Anomaly (PA) is linked to the Siberian Traps is NOT the subject of our paper, even if we discuss this possibility in the manuscript and in the *Supplementary Information* since it had been proposed in the literature in the past. Furthermore, in our research and findings, it is not relevant whether the plume is associated with the Siberian, Emeishan, or Icelandic LIPs. What matters is the existence of an upwelling in the geological past, which led to the heating of the lithosphere as it moved over the upwelling. Therefore, we have proposed that, in the case of an *existing* mantle plume (whatever its origin may be), the influence of this plume on the development of the sedimentary basins during the period of its thermal decay was profound in terms of tectonic and thermal evolution.

The link between the Siberian traps and the PA was proposed by Lekic et al. (2012) based the restored location of the eruption of the Siberian traps (Torsvik et al., 2008), which is close to the projection of the PA. In addition, there is another hypothesis proposed by Flament et al. (2017, Nat. Commun.) that the PA could be linked to the Emeishan LIP. However, Torsvik & Domeier (2017, Nat. Commun.) dismissed this hypothesis considering (i) the stability of large low-shear-velocity provinces (LLSVP), (ii) the location of the Emeishan LIP eruption site near the equator 260 My ago, and (iii) its association with the JASON and not the TUZO LLSVP.

Finally, we do not discuss the origin of the PA anomaly per se, as it is not a subject of our paper; that is why, we placed the discussion of this topic in the *Supplementary Information* (see SD1), as supporting the main text but not a key issue for our paper.

Problems with this scenario include (i) that no plume is imaged above the Perm Anomaly at present-day, (ii) that there is no volcanism in the East European platform, (iii) that the link between the Perm Anomaly and the Siberian Traps is not demonstrated in the manuscript, and (iv) that deep basal mantle structures may have been mobile over time.

- (i) According to analysis of seismic tomographic images (e.g., Fig. 1), the PA is likely to be a remnant part of potentially existing mantle plume as it was suggested by the authors mentioned above. If a plume is seismically imaged above the PA, the current manuscript would have no sense at all, as the interaction of a mantle plume with the lithosphere and impacts of prominent plumes on sedimentary basin evolution are well-studied scientific topics (e.g., let us just mention a classical paper on these topics by McKenzie, 1978). **The new idea presented in our manuscript is that a weakened (but long survived) heat source associated with a thermally decaying plume (and hence seismically**

undetected) can influence the thermal condition of the lithosphere and basins, below which it was located during a long geological time (some 150-200 My).

- (ii) If the PA is a remnant of a mantle plume linked to the Siberian LIP, major magmatic events were associated with the P-T boundary (Siberian traps). In the beginning of the Early Triassic, magmatism was still active in West Siberia (Nikishin et al., 2002). In the Late Triassic (225-207 My ago) magmatism was manifested as tholeiitic and calc-alkali basalt eruptions in the Turgay region to the southeast of the Urals (Bochkarev et al., 2000; Bochkarev, 2001). There have been no substantial indications of volcanism recorded in the EE lithosphere since the Early Jurassic times. One of the reasons for that is that the EE lithosphere was thick enough to allow for magmatic eruptions. The thickness of the lithosphere exceeds 150 km in the studied area: this is visible in the S-wave velocity images of the EE lithosphere (see Fig. S3 of the *Supplementary Information*) plotted using the data from Shapiro & Ritzwoller (2002), and in the lithosphere-asthenosphere boundary maps based on the data from Hoggard et al. (2020) (Fig. S4 of the *Supplementary Information*) and from Artemieva (2006) (Fig. S5 of the *Supplementary Information*). According to Davis et al. (2015), the absence of magmatic events in the studied region of the EE platform can be explained by the thick (>150 km) lithosphere. A relevant discussion is included in SD2 of the *Supplementary Information*. Also we revised the main text discussion briefly the points (see lines 186-194 of the revised manuscript with revision marked)
- (iii) The aim of our paper is not to explain and prove the link between the Perm Anomaly and the Siberian Traps, as it has already been done in a series of papers, one of which has been published in *Nat. Commun.* (Torsvik & Domeier, 2017). Please see more on this topic above.
- (iv) We agree with Reviewer 3 that there are no structures immobile over a long time in the convecting in the Earth's mantle. Still, there are two basic hypotheses related to the stability and mobility of mantle structures (e.g. plumes) based on two end-members views, which may each be supported by chosen geological, geophysical or geochemical observations. One of the hypotheses is that the mantle plumes remain fixed in space, and another hypothesis is that the plumes are mobile deep structures. Of course, the reality is more complicated than any of the end-member hypotheses. Here, we assume that the PA structure remained reasonably fixed in space for at least 300 My based on the subducted lithosphere surrounding the PA (see Fig. 1).

The scenario presented in Figure 2 suggests a solution to problem (i) in which the lifetime of the plume tail would be limited to 50 million years or less (specifically, Figure 2a,b,c). If this is the case, why are so many plume tails long-lived as demonstrated by oceanic hotspot tracks that span about 100 million years (see for example O'Neill et al., 2005)?

The exact time at which we see the tail disappearing depends on the isotherm that is being considered; a colder isotherm would disappear later. However, whatever the isotherms, the plume conduit will eventually be cut, and laboratory experiments are showing this effect on both the temperature and the velocity fields (Davaille & Vatteville, 2005). Moreover, they show that within 60% of its lifetime, the velocity in most of the plume conduit would have already decayed to 30% of its overall maximum (reached during the «head» event). For a typical lower mantle viscosity, the total duration of a plume would be about 200-300 My (e.g. Davaille & Vatteville, 2005; Ismail-Zadeh et al, 2006), with about 100-150 My from a plume initiation to its mature phase, which is characterized by a prominent plume tail throughout the mantle. After that, it takes additional 50-100 My for the plume tail to thermally decay and to disappear. Hence, plumes may remain active for > 150 My generating oceanic hotspots tracks. Anyway, we have modified panels of Fig. 2 (b-f), considering the numerical results obtained at a higher Rayleigh number closer to the lab experiment of Fig. 2a to make clearer the effects of the thermal diffusion of the advected mantle plume.

As opposed to plume tails, plume heads are thought to be short-lived and to result in the eruption of large igneous provinces over 1 to 5 million years (Bryan and Ernst, 2008). In contrast, in Figure 2, the plume head persists for at least 200 million years. Is the model presented in Figure 2 applicable to Earth?

Bryan, Scott E., and Richard E. Ernst. "Revised definition of large igneous provinces (LIPs)." *Earth-Science Reviews* 86.1-4 (2008): 175-202.

The numerical model of the mantle plume evolution presented in the manuscript does not consider melting, melt accumulation beneath the lithosphere, melt penetration through the lithosphere, volcanic

eruptions, lava flows, and a development of lava provinces. If the complications have been introduced in the model, we may expect the plume head to be short-lived. As melting and eruptions are not topics of our research, we do not introduce these complications into the numerical model because this complicated model will not significantly enhance our numerical results related to diffusion of mantle plumes. The proposed simple numerical model explains the disappearing tail and the diminishing head with respect to the plume foot without melting and eruption accounted. We have added a relevant discussion into the manuscript (see lines 119-125 of the revised main text with the revision marked).

About the applicability of the model to the Earth, we note that this model is based on the essential equations used in fluid mechanics, thermodynamics, and mantle convection (Ismail-Zadeh and Tackley, 2010). No model can describe the dynamics of the Earth's mantle in all detail but the dynamic processes in the planet can be identified in the framework of a considered model. The simpler model is (i.e., the fewer fitting parameters are used), the easier is an interpretation and the better is understanding of numerical results. If a simple model can explain the reality, we do not need to complicate the model to explain the features which are not of research interest. "With four exponents I can fit an elephant" told Enrico Fermi on this case. A model introducing many complications is useful when it fits research goals. In our case, the most important was to show the evolution of a fading mantle plume, and our simple model can do it.

Regarding problem (ii), it is argued that the plume was waning and that the East European lithosphere is thick, preventing melts from reaching the surface (l. 131-134). Davies et al. (2015) documented a long hotspot track beneath the eastern part of the Australian continent, and they showed that melting occurred where the lithosphere was less than 150 km thick. It is unclear that the thickness of the Eastern European lithosphere is more than 150 km in the eastern part of the region of interest (see for example Steinberger and Becker, 2018 or the global lithospheric thickness models available from Hoggard et al., 2020 at <https://osf.io/twksd/>).

Davies, D., Rawlinson, N., Iaffaldano, G. et al. Lithospheric controls on magma composition along Earth's longest continental hotspot track. *Nature* 525, 511–514 (2015).

Hoggard, Mark J., et al. "Global distribution of sediment-hosted metals controlled by craton edge stability." *Nature Geoscience* 13.7 (2020): 504-510.

Steinberger, Bernhard, and Thorsten W. Becker. "A comparison of lithospheric thickness models." *Tectonophysics* 746 (2018): 325-338.

We thank you for the comment resulted in an analysis of the thickness of the EE lithosphere based on seismic (Shapiro & Ritzwoller, 2002; Steinberger & Becker, 2018; Hoggard et al., 2020) and heat flow data (Artemieva, 2006). The analysis has shown that the thickness of the lithosphere exceeds 150 km in the studied area. Please see Fig. S3 (based on data from Shapiro & Ritzwoller, 2002), Fig. S4 (based on the data from Hoggard et al., 2020; thanks for the link to the data), and Fig. S5 (based on data from Artemieva, 2006) in the *SI*. Hence, according to Davis et al. (2015), the long magmatic gap observed in the studied region of the EE platform since the Early Jurassic times can be explained by the thick (>150 km) lithosphere. The relevant discussion has been introduced (see lines 187-188, 192-194 of the revised manuscript with the revision marked) and in the *Supplementary Information* (see SD2).

The link between the Perm Anomaly and the Siberian Traps is proposed based on a reconstruction from 1998 (Supplementary Figure 1) based on which it is argued that 'the EE platform moved north-eastward at the rate of about (5°N, 10°E) in My in the Middle to Late Triassic times.' (l. 83-84). It would be helpful to quote a rate in degrees per million years.

This is replaced by 6 cm yr⁻¹.

It would also be helpful to link the Perm Anomaly to the Siberian Traps as in Figure 3 of the main text (which links the Perm Anomaly to the Eastern European platform) by using a tectonic reconstruction that extends at least back to 250 million years ago (examples include the reconstructions of Matthews et al., 2016 or Young et al., 2019).

Young, Alexander, et al. "Global kinematics of tectonic plates and subduction zones since the late Paleozoic Era." *Geoscience Frontiers* 10.3 (2019): 989-1013.

The reconstruction of the PA's position relative to Siberia using APM by Matthews et al. (2016) is now presented in Fig. 3 of the manuscript and in Fig. S6 of the *Supplementary Information*. Their reconstruction is similar to that by Kazmin and Natapov (1998) mentioned in the *Supplementary Information* (SD1). According to this model, the PA was located to the south of the West Siberian Basin between 240 My and 260 My and likely to be responsible for the Permo-Triassic volcanism found in drilling and attributed to a plume by Saunders et al. (2004).

In the rebuttal, it is argued that the focus of the study is the past 200 million years, however, in the absence of geophysical evidence for a plume and of any volcanism linked to the Perm Anomaly in the last 250 million years, it seems reasonable to request some geological evidence linking the Perm Anomaly to a mantle plume.

Geophysical (seismological) evidence can illustrate only the present state of the mantle (everything beyond it is interpretations). We are studying a fading mantle plume which is likely imaged by seismic tomography at the core-mantle boundary (the PA). As for the link between a mantle plume and the PA, this was discussed earlier and the link was proposed earlier by Lekic et al. (2012), Flament et al. (2017), and Torsvik & Domeier (2017) based on seismic and geodynamics studies. Paleogeographic and palinspastic reconstructions based on paleomagnetism (Kazmin & Natapov, 1998) and true polar wander-corrected paleomagnetic reconstructions (Torsvik et al., 2006; Matthews et al., 2016) provide another evidence of a possible connection between the PA and the Siberian trap province.

If the Perm Anomaly is linked to the Siberian Traps, should there be some volcanism between the Siberian Traps and the East European platform in places where the lithosphere is less than 150 km thick?

If the PA is linked to the Siberian traps, there is indeed evidence of volcanism in the West Siberian basin, where the seismic thickness of the lithosphere is likely to be less than 150 km (see, e.g. Fig. S1 of the *Supplementary Information*). See also our response about the magmatism between 250 My and 200 My ago in the West Siberia. A relevant discussion has been added in the *Supplementary Information* (SD2).

While it would be easier to link deep seismically slow anomalies to past volcanic eruptions if the structure of the deep Earth was rigid and fixed over time, Flament et al. (2022) showed that volcanic eruptions could be linked to either fixed or mobile deep mantle structures.

Flament, Nicolas, et al. "Assembly of the basal mantle structure beneath Africa." *Nature* 603.7903 (2022): 846-851.

Thank you for the comment and reference to this interesting work. A discussion on this topic is added to SD1 of the *Supplementary Information*.

Sincerely,

Alik Ismail-Zadeh, Anne Davaille, Jean Besse, and Yuri Volozh

References

- Artemieva, I.M. Global $1^{\circ} \times 1^{\circ}$ thermal model TC1 for the continental lithosphere: Implications for lithosphere secular evolution. *Tectonophysics* **416**, 245–277 (2006).
- Bochkarev, V. S. Triassic volcanogenic rocks of the western Siberia. In: *Triassic West Siberia*, Novosibirsk, SNIIGGMS, pp. 70-79 (2001).
- Bochkarev, V. S., Brekhunsov, A. M. & Deschenya, N. P. Cardinal problems of stratigraphy of the Mesozoic oil-and-gas-bearing deposits. *Geology of Oil and Gas* **1**, 2-13 (2000).
- Davaille, A. & Vatteville, J. On the transient nature of mantle plumes. *Geophys. Res. Lett.* **32**, L14309, doi:10.1029/2005GL023029 (2005).
- Davies, D., Rawlinson, N., Iaffaldano, G. et al. Lithospheric controls on magma composition along Earth's longest continental hotspot track. *Nature* **525**, 511–514 (2015).

- Flament, N., Williams, S., Muller, R.D., Gurnis, M. & Bower, D.J. Origin and evolution of the deep thermochemical structure beneath Eurasia. *Nature Communication* **8**, 14164, DOI: 10.1038/ncomms14164 (2017).
- Hoggard, M.J., Czarnota, K., Richards, F.D. et al. Global distribution of sediment-hosted metals controlled by craton edge stability. *Nat. Geosci.* **13**, 504–510 (2020).
- Ismail-Zadeh, A., Schubert, G., Tsepelev, I. & Korotkii, A. Three-dimensional forward and backward numerical modeling of mantle plume evolution: Effects of thermal diffusion. *J. Geophys. Res.* **111**, B06401, doi:10.1029/2005JB003782 (2006).
- Ismail-Zadeh, A., and Tackley, P.J. *Computational Methods for Geodynamcs*, Cambridge University Press, Cambridge (2010).
- Ismail-Zadeh, A., Castelli, F., Jones, D. & Sanchez, S. *Applications of Data Assimilation and Inverse Problems in the Earth Sciences*. Cambridge University Press, Cambridge (2023).
- Kazmin, V.G. & Natapov, L.M. The Paleogeographic Atlas of Northern Eurasia. 26 maps: Devonian to Neogene (380.8–6.7 Ma). TOO Institute of Tectonics of Lithospheric Plates, Moscow (1998).
- Lekic, V., Cottaar, S., Dziewonski, A. & Romanowicz B. Cluster analysis of global lower mantle tomography: A new class of structure and implications for chemical heterogeneity. *Earth Planet. Sci. Lett.* **357–358**, 68–77 (2012).
- Matthews, K.J., Maloney, K.T., Zahirovic, S., Williams, S.E., Seton, M. & Müller, R.D. Global plate boundary evolution and kinematics since the late Paleozoic. *Global and Planetary Change* **146**, 226–250 (2016).
- McKenzie, D. Some remarks on the development of sedimentary basins. *Earth Planet. Sci. Lett.* **40**, 25–32 (1978).
- Nikishin, A.M., Ziegler, P.A., Abbott, D., Brunet, M.-F. & Cloetingh, S. Permo-Triassic intraplate magmatism and rifting in Eurasia: implications for mantle plumes and mantle dynamics. *Tectonophysics* **351**, 3–39 (2002).
- O'Neill, C., Müller, D. & Steinberger, B. On the uncertainties in hot spot reconstructions and the significance of moving hot spot reference frames. *Geochem. Geophys. Geosyst.* **6**, Q04003, doi:10.1029/2004GC000784 (2005).
- Saunders, A.D., England, R.W., Reichow, M.K. & White, R.W. A mantle plume origin for the Siberian traps: uplift and extension in the West Siberian Basin, Russia. *Lithos* **79**, 407–424 (2005).
- Shapiro, N.M. & Ritzwoller, M.H. Monte-Carlo inversion for a global shear velocity model of the crust and upper mantle, *Geophys. J. Int.* **151**, 88–105 (2002).
- Steinberger, B. & Becker, T.W. A comparison of lithospheric thickness models. *Tectonophysics* **746**, 325–338 (2018).
- Torsvik, T.H. & Domeier, M. Correspondence: Numerical modelling of the PERM anomaly and the Emeishan large igneous province. *Nature Communication* **8**, 821, <https://doi.org/10.1038/s41467-017-00125-2> (2017).
- Torsvik, T.H., Smethurst, M.A., Burke, K. & Steinberger, B. Large igneous provinces generated from the margins of the large low-velocity provinces in the deep mantle. *Geophys. J. Inter.* **167**, 1447–1460 (2006).
- Torsvik, T.H., Steinberger, B., Cock, L.R.M. & Burke, K. Longitude: Linking Earth's ancient surface to its deep interior. *Earth Planet. Sci. Lett.* **276**, 273–282 (2008).

REVIEWERS' COMMENTS

Reviewer #2 (Remarks to the Author):

The authors have satisfactorily addressed my comments/concerns. I am attaching an annotated manuscript with a few minor editorial suggestions (wording/grammar).

Reviewer #3 (Remarks to the Author):

I thank the authors for carefully considering my comments. I accept that if the Perm Anomaly was ever associated with any mantle plume, that mantle plume has not (yet) been imaged. There is therefore room for the proposed scenario, especially because the East European lithosphere is thick, which would have limited surface volcanic eruptions.

It would be helpful to clarify how applicable the experiment is to Earth in general and to the region of interest in particular. Figure two juxtaposes paleogeographic reconstructions with a plume model. One problem is that there are (as far as I can tell) no subduction zones or slabs in the model, but the history of subduction to the north of the Neo Tethys and to the East of Asia is long and complex (this is clear from Fig. S1). Also, how comparable is a regional cartesian 3D plume model to this region of Earth (slabs and plumes, spherical) over tens of millions of years?

Some discussion was provided in the reply to reviewer comments, however, the discussion in the manuscript is limited to the implications of the model in terms of hydrocarbon generation. It would be helpful to discuss the merits and limitations of the model in the main text so that all readers (specialists and non-specialists) can appreciate them. It seems to me that much of the discussion is currently in the supplement (SD1. On the Perm Anomaly origin, SD2. Magmatism of the East European platform since the Triassic times, and SD5. Mantle plume diffusion modelling: Limitations, uncertainties, and sources of errors). A discussion of the applicability of the (slab-free?) model to the region of interest would also be helpful.

The presented experiment has been changed (it is based on a different Rayleigh number). Should more than one model be presented, and a sensitivity study be carried out?

In the rebuttal, it is stated that 'it is not relevant whether the plume is associated with the Siberian, Emeishan, or Icelandic LIPs.' I am not convinced as this has implications for the past motion of the plate carrying the LIP and/or the source of the plume (the Perm Anomaly).

Reviewer #4 (Remarks to the Author):

The paper addresses possible connections between a decaying mantle plume below a thick continental lithosphere, the respective paleo plate configuration and related thermal and subsidence imprints on the intracontinental basins located on this lithosphere. I consider the connection made between the thermal

history recorded in the fill of intracontinental sedimentary basins as providing constraints for the plate-scale geodynamics at work during their deposition to be innovative. The mantle modelling part and the conductive cooling considerations are not new per se, but the integrated discussion with the basin record is new and links two traditionally separate scientific communities: the geodynamicists with the basin history analysts in search for hydrocarbons.

The authors connect the time frame of a slowly “disappearing” plume or in other words, a decaying thermal anomaly in the mantle beneath a thick and partly cratonic lithosphere to reduced subsidence rates or even uplift phases in the sedimentary basins on this continental lithosphere that nevertheless experience a thermal pulse. Such reduced subsidence and uplift and coeval thermal events are indeed recorded over much of North central Europe’s basins, even in the regions bordering the EEP to the West. Hiatuses in Mid-Late Jurassic deposits are widespread in northern Poland, Germany, Denmark and even the North Sea. There, the distinctly thinner Phanerozoic lithosphere also records magmatic activity coeval with the hiatus for which no real explanation had been found and hypotheses didn’t go any further than “thermal updoming” (Underhill et al.). To find hydrocarbons it was sufficient to know that it was hot and shallow and not why this was the case, but many studies prove that this was the case. In summary it was hot and shallow inspite of a very thick lithosphere and what the authors do in this paper is to quantitatively check if a slowly decaying former mantle plume can do the job. They further check consistency with plate reconstructions which adds an additional integration level.

The analytical thermal conductive modelling part doesn’t add much to the conclusion and definitely represents a very conservative estimate of any plume-related thermal effects. The internal heating is neglected, but can compose almost half of the heat in thick intracontinental (or even cratonic) crust and heat transport is very different in 3D than in 1D, especially in basins of the type considered here where radiogenic thermal blanketing severely influences the thermal field and its evolution (Stephenson, Scheck-Wenderoth and Maystrenko, Cacace and Scheck-Wenderoth). In addition, the boundary conditions chosen (lines 428-430) strongly control the result, so the result of this exercise is more a fitting exercise than a real forward simulation considering the physics at work. On the other hand, one has to start from somewhere and the authors justify their choices and provide transparency on the latter. In the end they are able to give an estimate of how the influence of the plume must have been to reach the measured and preserved maturity of the organic matter in the basins in the right time frame and this represents a value in itself. That they are able to connect the two time frames and dynamics (mantle heat and heat transport in the vertically moving lithosphere) with each other and with plate motion reconstructions makes the paper worth to be published.

A few minor comments: I am sure the authors are aware of these works, but I would like to see a reference that can be short or partly given in the Supplementary on how far the findings of the authors are consistent with them as they addressed

- the plume issue in 3D and gave first ideas on how the thermal imprints of plumes on cratonic lithosphere would evolve: Burov, E. & Gerya, T. 2014: <https://doi.org/10.1038/nature13703> and Koptev et al., 2016 : <https://doi.org/10.1016/j.gsf.2015.11.002>
- The Jurassic plume effects on Northern Europe without much of a larger plate tectonic frame: Underhill, J.R., Partington, M.A., 1993. Jurassic thermal doming and deflation in the North Sea: implications of the sequence stratigraphic evidence. In: Parker, J.R. (Ed.), *Petroleum Geology of Northwest Europe: Proceedings of the 4th Conference*. The Geological Society, London, pp. 337– 345.
- thermal cooling and radiogenic heat contribution in intracontinental basins: Stephenson, 2009. *Nat. Geosci*; <https://doi.org/10.1038/NGEO479> or Cacace, M. & Scheck-Wenderoth, 2016 *JGR*:

<https://doi.org/10.1002/2015JB012682>

• hydrocarbons at plume affected passive continental margins: Anka et al: 2010 Marine and Petroleum Geology. <https://doi.org/10.1016/j.marpetgeo.2009.08.015>

Figure 2 normalized subsidence has no vertical scale and normalization info in Methods doesn't help to understand what is shown...

Data availability statement (line 432) doesn't say anything. Where will the data be made available?

There are still a few spelling errors in the manuscript and the SI

RESPONSE TO THE REVIEWERS' COMMENTS (the reviewers' comments are marked by red)

Reviewer #2 (Remarks to the Author):

The authors have satisfactorily addressed my comments/concerns. I am attaching an annotated manuscript with a few minor editorial suggestions (wording/grammar).

Many thanks for the suggestions. We have implemented them in the final manuscript.

Reviewer #3 (Remarks to the Author):

I thank the authors for carefully considering my comments. I accept that if the Perm Anomaly was ever associated with any mantle plume, that mantle plume has not (yet) been imaged. There is therefore room for the proposed scenario, especially because the East European lithosphere is thick, which would have limited surface volcanic eruptions.

We thank the Reviewer as well for careful dedication to the manuscript's review and for many constructive comments and suggestions. We would like to reiterate our comments related to the activity of the PA plume in the geological past and its present seismic imaging. We note that "if the Perm Anomaly was ever associated with any mantle plume" in the past, a prominent shape of the mantle plume could never be seismically imaged today, as the plume would have already thermally faded given large geological time lapse. This is what has become the foundation of our work.

It would be helpful to clarify how applicable the experiment is to Earth in general and to the region of interest in particular. Figure 2 juxtaposes paleogeographic reconstructions with a plume model. One problem is that there are (as far as I can tell) no subduction zones or slabs in the model, but the history of subduction to the north of the Neo Tethys and to the East of Asia is long and complex (this is clear from Fig. S1).

Both laboratory and numerical experiments presented in this manuscript are applicable to the Earth dynamics and particularly to thermal plume evolution in the mantle, as both experiments describe the fluid (mantle) thermodynamics and allow generating thermal convective patterns (plumes and slabs). Moreover, the aim of Fig. 2 was to present schematically a mantle plume evolution beneath the studied region, and hence, only the hot isothermal surface ($T=3000\text{K}$) obtained in the numerical model was presented in the figure. If an isothermal surface at $T=600\text{-}800\text{K}$, for example, were presented, it would show a descending slab. However, lithospheric slabs are not the subject of this study, and by imaging them, we would introduce unnecessarily complications into the figure. A clarification is added to the main text of the MS (lines 302303).

Furthermore, the suggestion to model the evolution of the mantle upwelling and the lithosphere subduction in the region is fine, although we note that this suggestion is not related to the topic of this manuscript and to the essential conclusion of this work and will be considered by the authors in a future study.

Also, how comparable is a regional cartesian 3D plume model to this region of Earth (slabs and plumes, spherical) over tens of millions of years?

We would like to refer the Reviewer to the work by O'Farrell et al. (2013; <https://doi.org/10.1093/gji/ggs053>) on the comparison of a spherical shell model and 3D Cartesian (plane-layer) model of mantle convection, and to their conclusion that for Rayleigh numbers of greater than 10^5 (as in our case), the mean temperatures between the spherical shell and 3D Cartesian models "roughly converge for solution domain size(s) of $3 \times 3 \times 1$ " (the same solution domain size as used in the numerical modelling of thermal plume decay presented in this work). A related discussion is added to the SI (SD5).

Some discussion was provided in the reply to reviewer comments, however, the discussion in the manuscript is limited to the implications of the model in terms of hydrocarbon generation. It would be helpful to discuss the merits and limitations of the model in the main text so that all readers (specialists and non-specialists) can appreciate them. It seems to me that much of the discussion is currently in the supplement (SD1. On the Perm Anomaly origin, SD2. Magmatism of the East European platform since the Triassic times, and SD5. Mantle plume diffusion modelling: Limitations, uncertainties, and sources of errors). A discussion of the applicability of the (slab-free?) model to the region of interest would also be helpful.

The volume of the main body of the paper is limited, and, therefore, all the information, which the authors considered to be not essentially important for the conclusion of the paper, has been moved to the SI. Also, we note that the numerical model of mantle plume diffusion is not a 'slab-free' model as explained above.

The presented experiment has been changed (it is based on a different Rayleigh number). Should more than one model be presented, and a sensitivity study be carried out?

Thank you for the comment. A relevant discussion is added to the SI (SD5).

In the rebuttal, it is stated that 'it is not relevant whether the plume is associated with the Siberian, Emeishan, or Icelandic LIPs.' I am not convinced as this has implications for the past motion of the plate carrying the LIP and/or the source of the plume (the Perm Anomaly).

Our study is not related to the origin of the mantle plume, which remnant is seismically imaged as the PA. We discussed several studies related to this topic in the SI (SD1). One of the goals of our study was to understand the thermal evolution of the mantle beneath the EE platform for the last 200 My assuming that the seismically modelled PA at the core-mantle boundary is a relic of the mantle plume, which was active for a longer period of the geological time. Therefore, we analysed the evolution of the PA plume and the plate movement over the PA for the last 200 My. That is the reason of why we do not insist on the primary geographic origin of the PA plume, although this plume could very well be located beneath Siberia in the Late Permian times, as proposed by several studies mentioned in SI (SD1).

Reviewer #4 (Remarks to the Author):

The paper addresses possible connections between a decaying mantle plume below a thick continental lithosphere, the respective paleo plate configuration and related thermal and subsidence imprints on the intracontinental basins located on this lithosphere. I consider the connection made between the thermal history recorded in the fill of intracontinental sedimentary basins as providing constraints for the plate-scale geodynamics at work during their deposition to be innovative. The mantle modelling part and the conductive cooling considerations are not new per se, but the integrated discussion with the basin record is new and links two traditionally separate scientific communities: the geodynamicists with the basin history analysts in search for hydrocarbons.

Thank you very much for the comments.

The authors connect the time frame of a slowly "disappearing" plume or in other words, a decaying thermal anomaly in the mantle beneath a thick and partly cratonic lithosphere to reduced subsidence rates or even uplift phases in the sedimentary basins on this continental lithosphere that nevertheless experience a thermal pulse. Such reduced subsidence and uplift and coeval thermal events are indeed recorded over much of North central Europe's basins, even in the regions bordering the EEP to the West. Hiatuses in Mid-Late Jurassic deposits are widespread in northern Poland, Germany, Denmark and even the North Sea. There, the distinctly thinner Phanerozoic lithosphere also records magmatic activity coeval with the hiatus for which no real explanation had been found and hypotheses didn't go any further than "thermal updoming" (Underhill et al.). To find hydrocarbons it was sufficient to know that it was hot and shallow and not why this was the

case, but many studies prove that this was the case. In summary it was hot and shallow inspite of a very thick lithosphere and what the authors do in this paper is to quantitatively check if a slowly decaying former mantle plume can do the job. They further check consistency with plate reconstructions which adds an additional integration level.

We appreciate the comment and have added some discussion related to the doming in the main body of the manuscript (lines 256-258).

The analytical thermal conductive modelling part doesn't add much to the conclusion and definitely represents a very conservative estimate of any plume-related thermal effects. The internal heating is neglected, but can compose almost half of the heat in thick intracontinental (or even cratonic) crust and heat transport is very different in 3D than in 1D, especially in basins of the type considered here where radiogenic thermal blanketing severely influences the thermal field and its evolution (Stephenson, Scheck-Wenderoth and Maystrenko, Cacace and Scheck-Wenderoth). In addition, the boundary conditions chosen (lines 428-430) strongly control the result, so the result of this exercise is more a fitting exercise than a real forward simulation considering the physics at work. On the other hand, one has to start from somewhere and the authors justify their choices and provide transparency on the latter. In the end they are able to give an estimate of how the influence of the plume must have been to reach the measured and preserved maturity of the organic matter in the basins in the right time frame and this represents a value in itself. That they are able to connect the two time frames and dynamics (mantle heat and heat transport in the vertically moving lithosphere) with each other and with plate motion reconstructions makes the paper worth to be published.

Thank you very much for the constructive comment. Although our aim was to analyse the specific contribution of a mantle plume to the heat transfer in the lithosphere, we have revised the model of the conductive heat transfer to include the effect of internal heating in the lithosphere. Namely, we introduce another geotherm as the initial thermal state of the lithosphere before it came to the contact with a mantle plume. The geotherm (as explained now in the Methods) was modelled for the surface heat flux 45 mW m⁻² accounting for the temperature- and pressure-dependent thermal conductivity and for a heat production within the lower crustal rocks of 0.4 μW m⁻³. The results of the modelling have changed, but they have not changed the conclusion of the work. The manuscript has been revised accordingly (lines 171-180, 359-383).

A few minor comments: I am sure the authors are aware of these works, but I would like to see a reference that can be short or partly given in the Supplementary on how far the findings of the authors are consistent with them as they addressed:

- the plume issue in 3D and gave first ideas on how the thermal imprints of plumes on cratonic lithosphere would evolve: Burov, E. & Gerya, T. 2014: <https://doi.org/10.1038/nature13703> and Koptev et al., 2016 : <https://doi.org/10.1016/j.gsf.2015.11.002>
- The Jurassic plume effects on Northern Europe without much of a larger plate tectonic frame: Underhill, J.R., Partington, M.A., 1993. Jurassic thermal doming and deflation in the North Sea: implications of the sequence stratigraphic evidence. In: Parker, J.R. (Ed.), *Petroleum Geology of Northwest Europe: Proceedings of the 4th Conference*. The Geological Society, London, pp. 337– 345.
- thermal cooling and radiogenic heat contribution in intracontinental basins: Stephenson, 2009. *Nat. Geosci*; <https://doi.org/10.1038/NCEO479> or Cacace, M. & Scheck-Wenderoth, 2016 *JGR*: <https://doi.org/10.1002/2015JB012682>
- hydrocarbons at plume affected passive continental margins: Anka et al: 2010 *Marine and Petroleum Geology*. <https://doi.org/10.1016/j.marpetgeo.2009.08.015>

Discussion and some references have been added to the SI (SD5).

Figure 2 normalized subsidence has no vertical scale and normalization info in Methods doesn't help to understand what is shown...

The vertical scale is not needed, as the curves are normalized, and their vertical size ranges from 0 to 1. The normalisation procedure is rewritten in the manuscript to clarify the procedure (lines 348-356).

Data availability statement (line 432) doesn't say anything. Where will the data be made available?

Data will be available in a 'github' site before the paper is published. It is now mentioned in the manuscript.

There are still a few spelling errors in the manuscript and the SI.

We have fixed them.

We thank the reviewer for the constructive comments.